# LAYERIF: Estimating Layer Quality for Large Language Models using Influence Functions

**Hadi Askari[†], Shivanshu Gupta[§], Fei Wang [‖], Anshuman Chhabra[†‡] Muhao Chen[†]**
[†] University of California, Davis [§] University of California, Irvine
[‖] University of Southern California [††]University of South Florida

## Abstract

Pretrained Large Language Models (LLMs) achieve strong performance across a wide range of tasks, yet exhibit substantial variability in the various layers' training quality with respect to specific downstream applications, limiting their downstream performance. It is therefore critical to estimate layer-wise training quality in a manner that accounts for both model architecture and training data. However, existing approaches predominantly rely on model-centric heuristics (such as spectral statistics, outlier detection, or uniform allocation) while overlooking the influence of data. To address these limitations, we propose **LAYERIF**, a data-driven framework that leverages *Influence Functions* to quantify the training quality of individual layers in a principled and task-sensitive manner. By isolating each layer's gradients and measuring the sensitivity of the validation loss to training examples by computing layer-wise influences, we derive data-driven estimates of layer importance. Notably, our method produces *task-specific* layer importance estimates for the *same* LLM, revealing how layers specialize for different test-time evaluation tasks. We demonstrate the utility of our scores by leveraging them for two downstream applications: (a) expert allocation in LoRA-MoE architectures and (b) layer-wise sparsity distribution for LLM pruning. Experiments across multiple LLM architectures demonstrate that our model-agnostic, influence-guided allocation leads to consistent gains in task performance.

## 1 Introduction

Deep neural networks, particularly Large Language Models (LLMs), have become increasingly powerful, achieving impressive performance across numerous natural language processing tasks such as question answering [1, 2], natural language inference [3], commonsense reasoning [4, 5], and code generation [6]. Recent models have demonstrated strong generalization across these benchmarks, such as Mistral [7], Gemma [8], GPT-4 [9], DeepSeek [10], LLaMA [11], and Qwen [12] model families. However, despite their effectiveness, these models exhibit significant internal variability, with layers differing substantially in terms of training quality and contribution to overall model performance [13–16]. Recent diagnostic tools, such as layer-wise adaptive learning rates [17, 18] and pruning strategies [19] informed by empirical spectral densities (ESDs), have highlighted that not all layers within a deep model are equally well-trained or consequential to its final performance.

There is a broad effort in the ML community to interpret and explain the behavior of these complex models. Many approaches take a model-centric perspective, examining how the model's learned weights effect predictions [20, 19]. Other methods use activations, or attention patterns to deduce internal mechanisms [21–24]. In contrast *Influence Functions (IFs)* [25, 26] offer a more explicitly data-centric perspective by quantifying the effect of individual training points on model's loss landscape. Rather than focusing on internal representations, Influence Functions (IFs) measure how a model's output would change if a given training example was perturbed or removed, effectively

39th Conference on Neural Information Processing Systems (NeurIPS 2025).

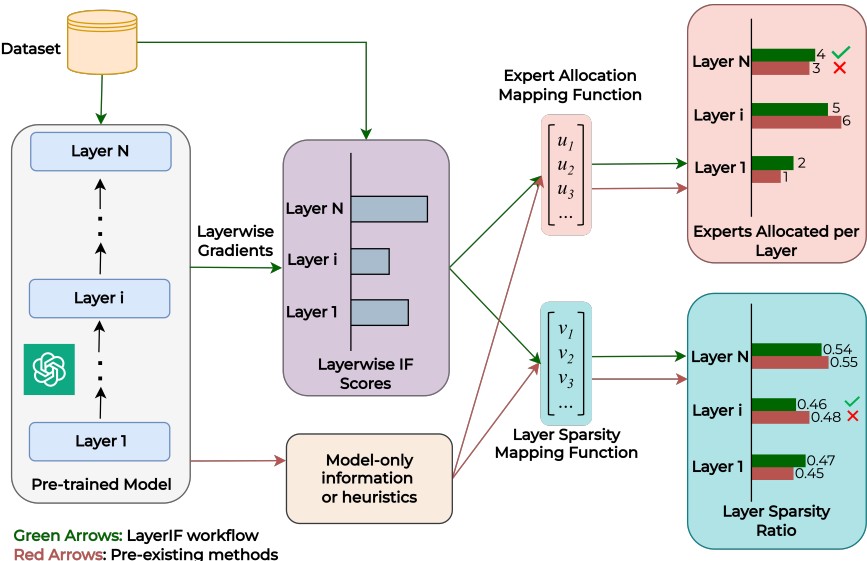

Figure 1: We present an overview of the LAYERIF pipeline via the green arrows. We quantify per-layer quality in a pretrained LLM using Influence Functions (IFs), demonstrating that the same model produces distinct layer-wise IF scores across different datasets, revealing dataset-specific specialization. The pipeline begins by extracting gradients at each Transformer block of the LLM, computed separately for each dataset. These gradients and the dataset are then used to estimate layer-wise IF scores that serve as data-driven proxies for layer quality. To demonstrate the utility of these scores, we consider two downstream applications: (a) the allocation of optimal experts per layer in a LoRA-MoE architecture, and (b) the computation of structured layer-wise sparsity ratios for model pruning. For each task, we apply a dedicated mapping function, to transform the raw IF scores into task-specific layer importance measures. On the contrary, pre-existing methods only use model-only information or heuristics to compute these metrics as indicated by the red arrows. Figure 1 is a conceptual illustration contrasting our method with the baselines; the shown values are illustrative, not experimental.

tracing the influence of training data on the model's predictions. They have proven effective in a range of downstream machine learning tasks, including mislabeled data detection [27–29], optimal subset selection [30–32], model interpretation [33–35], data attribution [36], data valuation [37], in-context learning [38], and analyzing model biases [39, 40].

Since IFs offer an empirical approach to measure the impact of training samples on model predictions, they present a promising avenue for assessing and quantifying this variability in layer quality. Surprisingly, despite their clear potential, Influence Functions have not yet been formally applied or systematically formulated for evaluating the quality or importance of individual layers within deep neural networks, particularly LLMs. In this work, we address this gap by introducing a novel framework, LAYERIF, that leverages Influence Functions to rigorously quantify and interpret the training efficacy of individual model layers. By systematically analyzing each layer's gradient information through IFs, we can obtain actionable insights into layer importance.

A key advantage of using IFs for estimating layer quality is their ability to combine perspectives from both the data- and model-level. For instance, the same model (if trained to different number of epochs) may exhibit layer-wise specialization that varies depending on the dataset or task, and IFs can capture this variability. Specifically, as part of our LAYERIF framework (illustrated in Fig. 1), we hypothesize that since IFs measure the sensitivity of model loss to individual training points and the loss of well-trained layers should show smaller changes, layer-wise IF scores can serve as a proxy for layer quality. These task-specific layer quality estimates can then be mapped to downstream settings using corresponding transformation functions.

We evaluate our LAYERIF framework on two practical downstream settings: (a) expert allocation in Mixture-of-Experts (MoE) architectures, and (b) layer sparsity allocation for model pruning. In recent work, MoE models have emerged as a compelling strategy for scaling LLMs efficiently by activating only a small subset of specialized subnetworks ("experts") per input token [41, 42].

While several parameter-efficient fine-tuning methods (e.g., LoRA) have been extended to MoE settings by training multiple low-rank adapters (LoRA experts) with learned routing mechanisms [43–45], typically, the same number of experts are allocated to every Transformer block. Recent studies [46, 47] have questioned this uniform allocation, finding that expert redundancy and routing overfitting can degrade performance. However, the alternatives proposed by them rely on group-based heuristics or model-driven ablations. In contrast, we propose to use *data-* and *model-driven* IF-based layer quality estimates to inform expert allocation.

Similarly, for the setting of layer sparsity allocation for LLM pruning, the traditional approach of imposing a uniform sparsity ratio per layer limits the ability to push toward greater global sparsity without sacrificing performance [48, 49]. Recent work on more principled approaches for allocating sparsity across layers includes OWL [20] and AlphaPruning [19], which perform non-uniform sparsity allocation using metrics derived from weight statistics (e.g., outlier activations or heavy-tailed spectral properties). These approaches rely solely on model weights and heuristics, without leveraging training data. Our method offers a complementary, data-driven alternative: we use LAYERIF to identify layers with lower aggregate influence and prune them more aggressively, while preserving better trained layers. In both pruning and MoE routing, our results show that LAYERIF-informed decisions yield improved accuracy trade-offs over uniform, heuristic, or solely model-driven baselines. Moreover, our method applies to both pretrained and fine-tuned models and is agnostic to the model architecture.

**Contributions.** In sum, our work advances layer quality estimation in LLMs through the following key contributions and findings:

- To the best of our knowledge, we are the first work to show that Influence Functions (IFs) can be effectively used to analyze LLM layer quality using our LAYERIF framework.
- We study the effectiveness of LAYERIF in two downstream tasks: **(a) expert allocation in Mixture-of-Experts (MoE) architectures**, and **(b) layer sparsity allocation for model pruning.**
- For expert allocation in Mixture-of-Experts (MoE) architectures we conduct experiments on Mistral-7b-v0.1 and Gemma-7b by computing the post fine-tuning accuracy over several GLUE (CoLA, MRPC) and QA (CommonsenseQ, OpenbookQ, TextScienceQ) datasets. This leads to percentage increase of 1.61% over the best performing baseline.
- For layer sparsity allocation for model pruninng we conduct experiments on Mistral-7b-v0.1 and Gemma-7b on several by computing the post-pruning zero shot accuracy on several NLP tasks namely BoolQ, Hellaswag, Winogrande, RTE, OpenbookQA, ARC Easy and ARC Challenge. This leads to a 0.90% increase over the next best baseline.

## 2 Related Works

**Layer Quality Estimation.** Prior work has examined the representational differences that emerge across layers within deep neural networks [50–52]. Several probing techniques have been deployed to capture the semantics of internal layer representations [53, 54]. Additionally, [55] employs random matrix theory to analyze the weight matrices of Deep Neural Networks. They show that these matrices exhibit heavy tail empirical spectral density (ESD) and a decay coefficient of this ESD, PL_Alpha_Hill, can effectively gauge layer quality. More recent work has used Shapley Values [56] to identify which layers contribute more to the model's capabilities [57]. Other research has found that the middle layers in LLMs provide stronger features for downstream tasks [58].

**Model Pruning and Layer-wise Sparsity Budgets.** Pruning is a long-established technique in neural network optimization, aimed at reducing the size of a trained model by eliminating redundant or non-essential parameters. [59, 60]. Modern works have used the similarity between the representations at different layers to identify the optimal layers to prune, finding that the deepest layers can be pruned with minimal degradation [61].[62] Compared the change in cosine similarity of the input and output representations of a layer to prune the model. Similarly, [63] prunes model structures based on gradient information, followed by LoRA finetuning, [48] prunes based on the Hessian inverse, and [49] uses weights and activations of layers to introduce sparsity. Computing layer-wise sparsity ratios has been commonly used in older pruning methods [64] and quantization approaches [65]. [19] uses the aforementioned PL_Alpha_Hill metric [55] to calculate layer-wise sparsity ratios for pruning. Finally, [20] proposes a non-uniform, layer-specific sparsity scheme informed by the distribution of outlier activations across layers.

**Allocating Mixture of Experts.** Research in combining LoRA based finetuning [66] and Mixture of Experts Models [41] to boost performance is an actively growing field. Several works have proposed uniform strategies for allocating experts in LoRA-MoE settings [67–71]. Other works have looked into heuristics-based group methods [46] and Heavy-Tailed Self-Regularization Theory [47] based adaptive layer-wise methods to allocate experts. Other research directions aim to improve the composability of LoRA for cross-task generalization [72].

**Influence Functions.** Classical influence function approaches seek to estimate the effect of training samples on the model's loss, either by leave-one-out retraining [26, 73–76] or through gradient-based approximation [27, 77, 78, 35, 79]. However, most of these methods are inapplicable for large language models due to computational inefficiency or convexity assumptions [80, 29]. More recently, several influence function methods have been proposed for LLMs, such as DataInf [81], Arnoldi iteration [82], alternative Hessian forms [34, 83], self-influence [84, 85], ensemble approaches [36, 86], as well as less performant but compute-efficient Hessian-free approaches [29, 28, 32, 87]. It is important to note that our LAYERIF framework is agnostic to the choice of influence function, so new advancements made in this area can be directly utilized as part of our layer quality estimation approach.

# 3 Proposed Approach

In this section we will first define the preliminaries and notation that we will use and then formally define our method.

## 3.1 Preliminaries and Notation

We clarify that in this work, the term layer refers specifically to a Transformer block, which comprises multiple weight matrices, including those associated with the attention mechanism and the mlp projection layers. We will now describe the preliminaries for Influence Functions and then for the two tasks we consider in this work: (a) allocating mixture of experts for LoRA finetuning and (b) layer sparsity allocation for model pruning.

**Influence Functions.** Let the training dataset be $D^{\text{train}} = \{z_i\}_{i=1}^{n}$ and the validation dataset be $D^{\mathcal{V}} = \{z_j^{\mathcal{V}}\}_{j=1}^{m}$, with $f_\theta$ representing a pretrained large language model with parameters $\theta \in \Theta$ trained using loss $\ell$. The Influence Function (IF) quantifies the sensitivity of model parameters to individual training examples [25, 26, 88]. Formally, it characterizes how the optimal parameter estimate $\theta^*$ changes when a specific training point is infinitesimally up-weighted in the empirical risk objective. Specifically, for a training point indexed by $k \in [n]$ and perturbed by infinitesimal weight $\epsilon \in \mathbb{R}$, we can obtain perturbed model parameters as $\theta^{(k)}(\epsilon)$. Then, assuming the loss is twice differentiable, [27] showed that the influence of the $k$-th training point on the learned parameters $\theta^*$ can be derived by taking the limit of the perturbed solution with respect to $\epsilon$, as $\epsilon \to 0$:

$$I_{\theta^*}(z_k) := \frac{d\theta^{(k)}}{d\epsilon}\bigg|_{\epsilon=0} = -H(\theta^*)^{-1}\nabla_\theta \ell(z_k, \theta),$$

Where $H(\theta)$ is the Hessian matrix. Subsequently, the influence of a training sample $z_i \in D^{\text{train}}$ on the validation loss becomes:

$$I(z_i) = -\sum_{j=1}^{m} \nabla_\theta \ell(z_j^{\mathcal{V}}, \theta)^\top H(\theta)^{-1} \nabla_\theta \ell(z_i, \theta).$$

The influence function $I(z_i)$ measures the effect of an individual training example on the validation loss. Specifically, it captures whether the sample $z_i$ has a beneficial or detrimental impact on the model's predictive performance. A larger negative/positive value of $I(z_i)$ indicates that the data point contributes to a reduction/increase in the loss, thereby serving as a beneficial/detrimental example for model optimization.

**LoRA-MoE.** In the Mixture-of-Experts (MoE) paradigm, each Transformer layer is augmented with $N$ parallel expert modules [42]. Given an input $\mathbf{x} \in \mathbb{R}^d$, the MoE layer computes the output as: $\mathbf{o} = \mathbf{W}_0\mathbf{x} + \sum_{i=1}^{N} G_i(\mathbf{x})E_i(\mathbf{x})$ where $\mathbf{W}_0$ is the pre-trained weight matrix, $E_i(\mathbf{x})$ denotes the

output of the $i$-th expert, and $G_i(\mathbf{x})$ is the routing probability for expert $i$, typically produced by a softmax over a trainable gating network: $G(\mathbf{x}) = \text{Softmax}(\mathbf{x}\mathbf{W}_g)$ with $\mathbf{W}_g$ being the gating matrix. In practical implementations, only the top-$K$ experts are selected to compute $E_i(\mathbf{x})$, and their outputs are aggregated with load-balancing losses to avoid expert collapse.

**Layer Sparsity Allocation for Model Pruning.** Given a pretrained Transformer-based language model $N$ with $L$ layers, we denote the set of prunable weights by $W$, and the model architecture by a tuple $D = (d_1, d_2, \ldots, d_L)$, where $d_i$ represents the number of prunable parameters in the $i$-th layer. The goal of layer-wise pruning is to remove a fraction of parameters in $W$ while maintaining model performance. Rather than pruning uniformly across layers, a strategy is adopted to allocate different pruning budgets based on *layer quality*.

We now define the process of selecting important layers based on IF scores via LAYERIF. We then discuss how we can use LAYERIF for (a) allocating a mixture of experts for LoRA finetuning and (b) layer sparsity allocation for model pruning.

## 3.2 LAYERIF: Estimating Layer Quality via Influence Functions

To assess the relative quality of different layers in the model, we localize IF computation to each layer $l$. We denote the per-layer parameter vectors as $\theta^{(l)}$ for each layer $l \in \{1, \ldots, N\}$. Let $\nabla_{\theta^{(l)}} \ell_i$ and $H^{(l)}(\theta)$ be the gradient and Hessian restricted to layer $l$. We define the layer-specific influence score of training point $z_i$ as:

$$I^{(l)}(z_i) = -\sum_{j=1}^{m} \nabla_{\theta^{(l)}} \ell(z_j^{\mathcal{V}}, \theta)^{\top} \left(H^{(l)}(\theta)\right)^{-1} \nabla_{\theta^{(l)}} \ell(z_i, \theta)$$

For each layer $l$, we aggregate the influence scores across the training set, considering only positively influential samples after a sign inversion [1]: $S^{(l)} = \sum_{i=1}^{n} \mathbb{I}[I^{(l)}(z_i) > 0] \cdot I^{(l)}(z_i)$ where $\mathbb{I}[\cdot]$ is the indicator function. This yields a vector $S \in \mathbb{R}^N$, where each element $S^{(l)}$ captures the cumulative positive influence of training data on validation performance through layer $l$.

**Justification.** We now provide some intuitive justification for why the LAYERIF framework can estimate layer quality with a high degree of accuracy. It is important to note that IF theory states that positive/negative influence scores for samples increase/decrease model loss and are hence, detrimental/beneficial to learning the task. In the same vein, to assess overall detriment/benefit to the end-task performance as a scalar, we can sum over all positive/negative influence sample scores. Moreover, by restricting influence to a particular combination of training samples and layers we can obtain these scalar influence contributions for these layers. Since we do this layer-wise influence computation in LAYERIF for different layers but keep the training samples fixed, we essentially isolate the impact of different layers on end-task performance. Thus, LAYERIF can serve as an effective approach for estimating layer quality in a data- and model-centric manner, unlike prior approaches. Layers with lower cumulative positive influence $S^{(l)}$ exhibit less sensitivity to training data, indicating greater stability and training maturity. Conversely, higher scores suggest under-trained or less generalizable layers. Finally, our layer quality scores obtained through LAYERIF can subsequently be used to guide pruning or resource allocation strategies that are dependent on accurate layer quality estimates, as we show next.

## 3.3 Layer-wise Expert Allocation using LAYERIF

We now discuss how we can use our LAYERIF method to calculate the optimal distribution of a fixed set of $N$ parallel expert modules. To inform expert allocation, we evaluate the *quality* of each layer using LAYERIF, which measures the sensitivity of model predictions to individual training samples. We compute the IF matrix for each of the Transformer layers in an LLM on a train and validation set. To obtain a scalar score $v_i$ for each layer, we aggregate its corresponding IF values while excluding any training samples with negative influence, i.e., those that decrease the model's loss, from the final sum.

To map LAYERIF scores to expert allocations, we first invert the raw scores obtained for each layer. Since our IF values are inverted to reflect loss sensitivity, we apply a sign inversion to ensure that layers

---

[1]This sign inversion is done since it is more intuitive that positive influential samples are beneficial to the model.

with higher (i.e., less negative) IF scores remain lower in magnitude after transformation. Formally, for each layer $i$, we compute $\tilde{v}_i = -v_i$ where $v_i = -S^{(l)}$. Next, we apply a power transformation to modulate the sharpness of the allocation. Specifically, we raise the inverted values to a hyperparameter-controlled exponent $\beta > 0$, yielding transformed scores $\hat{v}_i = \tilde{v}_i^\beta$. This exponent controls the dispersion of the allocation: higher values of $\beta$ amplify disparities between layers. We then scale the transformed values such that the total allocation, excluding a baseline of one expert per layer, matches the desired target sum $T$. Specifically, we compute $f_i = \frac{\hat{v}_i}{\sum_{j=1}^m \hat{v}_j} \cdot (T - m)$, where $m$ is the number of layers, and the floor of each fractional allocation $f_i$ is taken with a minimum of 1 added: $s_i = \lfloor f_i \rfloor + 1$. This guarantees that each layer receives at least one expert. Due to flooring, the sum $\sum_i s_i$ may be less than $T$. We compute the remaining allocation budget $r = T - \sum_i s_i$ and distribute these remaining units to the $r$ layers with the largest fractional remainders $f_i - \lfloor f_i \rfloor$, ensuring the total sum constraint is satisfied: $s_i \leftarrow s_i + 1$, for top $r$ layers with highest $(f_i - \lfloor f_i \rfloor)$. This procedure produces a final expert allocation vector $S = [s_1, s_2, \ldots, s_m]$ such that $\sum_i s_i = T$ and $s_i \geq 1$ for all $i$.

For each module $t$ in layer $i$, let $\mathbf{W}_0^{i,t} \in \mathbb{R}^{m \times n}$ denote the frozen pre-trained weight matrix. To construct the experts, we instantiate $s_i$ low-rank adaptation modules comprising trainable matrix pairs $\{\mathbf{A}_j^{i,t}, \mathbf{B}_j^{i,t}\}_{j=1}^{s_i}$, where $\mathbf{A}_j^{i,t} \in \mathbb{R}^{m \times r}$ is initialized with random Gaussian weights and $\mathbf{B}_j^{i,t} \in \mathbb{R}^{r \times n}$ is initialized to zero. Here, $r \ll \min(m, n)$ denotes the rank of the adaptation [66].

A router $S_j^{i,t}$, parameterized by a trainable weight matrix $W_r^{i,t} \in \mathbb{R}^{n \times N_j}$, dynamically assigns experts for a given input $x$. Following the MoLA framework [46], we adopt a top-$K$ routing strategy, where only the $K$ most relevant experts are selected for computation. To encourage balanced expert utilization, we additionally incorporate a load balancing loss at each layer [89]. The routing process is formally defined as $S_j^{i,t}(x) = \frac{\text{TopK}(\text{Softmax}(W_r^{i,t}x), K)_j}{\sum_{j=1}^K \text{TopK}(\text{Softmax}(W_r^{i,t}x), K)_j}$, where $S_j^{i,t}(x)$ denotes the normalized assignment weight for expert $j$. The final layer output $h^{i,t}$ is computed as: $h^{i,t} = W_0^{i,t}x + \sum_{j=1}^K S_j^{i,t}(x) A_j^{i,t} B_j^{i,t} x$, where $W_0^{i,t}$ is a pre-trained base weight matrix, and $A_j^{i,t}$, $B_j^{i,t}$ are low-rank matrices parameterizing the $j$-th expert. Thus, the output $h^{i,t}$ combines the base transformation of $x$ with the aggregated contributions of the top-$K$ experts, each scaled by its corresponding assignment probability $S_j^{i,t}(x)$. A step-by-step walkthrough of our algorithm is provided in Appendix C for greater clarity.

## 3.4 Layer Sparsity Allocation for Model Pruning via LAYERIF

We describe how our LAYERIF method can be applied to the problem of allocating layer sparsity budgets for model pruning [19, 90]. The motivating hypothesis is that the less well-trained a layer is, the more it can be pruned as opposed to pruning each layer uniformly. We compute the IF scores for each Transformer layer in an LLM on a train and validation set. These scores are aggregated by summing the positive influence contributions across all samples, resulting in a vector of layer quality scores $s = (s_1, s_2, \ldots, s_L)$, where each $s_i$ quantifies the total positive influence attributable to the $i$-th layer.

To ensure stable and meaningful sparsity allocation, we post-process the raw layer-wise IF scores before applying the mapping function. First, we take the absolute value of each score, and normalize these values to the range $[0, 1]$ using min-max normalization. This normalization step standardizes the range of IF magnitudes across layers. Next, we apply the Savitzky–Golay filter [91] to the normalized vector $\tilde{s} = (\tilde{s}_1, \tilde{s}_2, \ldots, \tilde{s}_L)$ to smooth local fluctuations and reduce noise while preserving the overall trend in layer quality.[2] The smoothed scores are then used as input to the sparsity allocation function $\phi$ described below. This two-step normalization and smoothing process improves robustness and consistency in sparsity assignment, particularly in cases where the raw influence scores vary erratically between adjacent layers.

To convert these quality scores into sparsity budgets, we build upon ideas in prior work [19]. First, we apply a normalized linear mapping $\phi : \mathbb{R}^L \to \mathbb{R}^L$, namely $\phi(\tilde{s})_i = \eta \left( \frac{\tilde{s}_i - \tilde{s}_{\min}}{\tilde{s}_{\max} - \tilde{s}_{\min}} (e_2 - e_1) + e_1 \right)$, where $s_{\min}$ and $s_{\max}$ are the minimum and maximum elements in the vector $s$, and $e_1, e_2$ are tunable parameters that control the minimum and maximum sparsity levels per layer. The scaling factor $\eta$ is

---

[2]Raw Influence Function (IF) scores at adjacent layers exhibit high-frequency noise due to stochastic Hessian approximations and mini-batch gradient estimates. This variance is well documented by [34], whose layer–token heatmaps show similar oscillations.

Table 1: Accuracy across datasets using ALPHALORA, MOLA and LAYERIF variants. We report scores under three IF strategies: Top 25% positively influential samples, all positively influential samples (+ve), and all samples (top performers in bold).

| Dataset | ALPHALORA | MOLA | | | LAYERIF | | |
|---|---|---|---|---|---|---|---|
| | | 2468 | 5555 | 8642 | +VE | TOP 25% | ALL |
| MRPC | 75.30 | 82.02 | 80.64 | 80.70 | 82.49 | 82.63 | 81.63 |
| Cola | 82.07 | 82.26 | 85.14 | 84.56 | 83.60 | 83.80 | 84.47 |
| Text Science Q | 78.52 | 64.65 | 72.75 | 76.39 | 75.04 | 79.59 | 77.61 |
| Common Q | 82.06 | 80.91 | 79.93 | 81.00 | 80.26 | 80.75 | 80.34 |
| Openbook Q | 84.60 | 64.00 | 83.80 | 82.20 | 87.40 | 84.60 | 85.20 |
| Average | 80.51 | 74.77 | 80.45 | 80.97 | **81.76** | **82.27** | **81.85** |

computed to satisfy the global sparsity constraint $S$, such that $\sum_{i=1}^{L} \phi(\tilde{s})_i \cdot d_i = S \cdot \sum_{i=1}^{L} d_i$ with $d_i$ denoting the number of prunable parameters in the $i$-th layer. This ensures that the total number of remaining parameters matches the target sparsity level. We then incorporate these layer-wise sparsity ratios in LLM pruning methods Magnitude [92], Wanda [49] and SparseGPT [48] and evaluate zero-shot capabilities.

# 4 Experimental Results

We delineate our experimental setup for experiments and analysis, and discuss our dataset details, models used to conduct the experiments, our methods and baselines we compare against.

**Datasets and Base LLMs.** For the layer-wise expert allocation in the LoRA-MoE application, we assess post-training zero-shot performance on two GLUE datasets and three commonsense reasoning tasks, following a protocol similar to that of Qing et al. [47]. The datasets used include: MRPC [93], CoLA [94], ScienceQA [95], CommonsenseQA [96], and OpenBookQA [97]. For layer sparsity allocation for model pruning we again assess the post-pruning zero-shot performance of our models on several NLP tasks. Namely, BoolQ [98], Hellaswag [99], Winogrande [100], ARC Easy and ARC Challenge [101], RTE [94], and OpenBookQA [97]. For both of the applications we conduct experiments on Mistral-7b-v0.1 [7] and Gemma-7b [8]. All of our experiments run on 8×NVIDIA RTX 6000 Ada GPUs. All code and reproducibility configurations are provided in Appendix I and we provide further information on statistical significance of our results in G.

**Method Protocol and Baseline Configurations.** For the layer-wise expert allocation application we train our models for 3 epochs and then compare zero-shot accuracy on the datasets using a protocol similar to [47]. We experimented with 3 LAYERIF configurations: using all samples LAYERIF (ALL), using only positively influential training samples LAYERIF (+VE) and using the top 25 % of the most influential training samples for our layer-wise summation: LAYERIF (TOP 25%).

Additionally, we compare against ALPHALORA [47], and 3 baselines from MOLA [46]: MOLA-2468, MOLA-5555 and MOLA-8642.

For layer sparsity allocation for model pruning, we apply the layer-wise sparsities derived from our LAYERIF method to traditional LLM pruning methods such as Magnitude [92], Wanda [49], and SparseGPT [48], and compare zero-shot performance. All of these methods originally have uniform layer-wise sparsity, which we use as a baseline for comparison. We also compare with state-of-the-art baselines ALPHAPRUNING [19] and OWL [20]. We prune our LLMs to 50% sparsity, since we can achieve strong accuracy while removing half of the model weights, hence maintaining the real-world utility of our method.

## 4.1 Results on Layer-wise Expert Allocation using LAYERIF

**All LAYERIF configurations outperform baselines.** Table 1 presents accuracy across datasets for expert allocation methods, including ALPHALORA, three variants of MOLA, and our proposed LAYERIF configurations. LAYERIF consistently outperforms prior approaches, achieving the highest average accuracy across datasets. Specifically, the best performing LAYERIF configuration had a relative percentage increase of 1.61% over the best performing baseline and the average of the

Table 2: Comparison of average accuracies across different pruning strategies (Magnitude, Wanda, SparseGPT) and layer-wise sparsity allocation methods at 50% pruning on Mistral-7b-v0.1 (top performers in bold).

| Method | Magnitude | Wanda | SparseGPT | Average |
|---|---|---|---|---|
| Uniform | 56.41 | 58.71 | 60.45 | 58.52 |
| AlphaPruning | 56.74 | 58.48 | 60.07 | 58.43 |
| OWL | 56.16 | 58.75 | 59.99 | 58.30 |
| LayerIF | **56.89** | **58.94** | **60.61** | **58.81** |

LayerIF configurations had a 3.52% percentage increase over the average of other strong baselines. These results demonstrate that IF-based layer quality estimation provides a more effective signal for expert allocation in LLMs.

**Consistently strong performance across varying number of experts.** In Figure 2, we provide average results for varying the number of experts, specifically 80, 160 and 224 experts on Mistral-7b-v0.1. MoLA(1234) and MoLA(46810) are introduced as additional baselines, following the most performant pattern as claimed in the MoLA paper. We can clearly see that LayerIF clearly outperforms the baselines in all three settings by having an average accuracy that is a 3.64% percent increase over the second best average accuracy AlphaLora. Expert allocation results for Gemma-7b are provided in Appendix D.

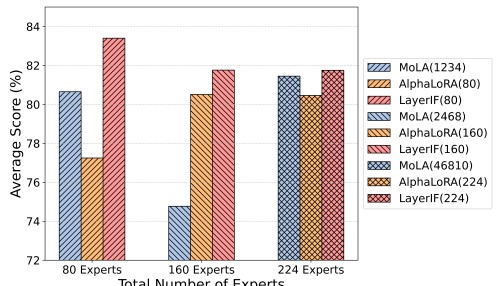

Figure 2: Comparison between AlphaLora, MoLA and LayerIF with varying number of total experts (80, 160, 224).

## 4.2 Results on Layer Sparsity Allocation for Model Pruning via LayerIF

**Superior performance on all pruning techniques.** Table 2 clearly indicates that using LayerIF for layer-wise sparsity allocation leads to performance gains over the baselines with a 1.03% percentage increase over OWL and a 0.90% increase over AlphaPruning in SparseGPT respectively. The results reiterate the value of IF-based layer quality estimation.

**Robust performance across multiple pruning ratios.** Figure 3 demonstrates the consistently reliable performance of LayerIF across multiple pruning ratios. LayerIF consistently achieved the highest

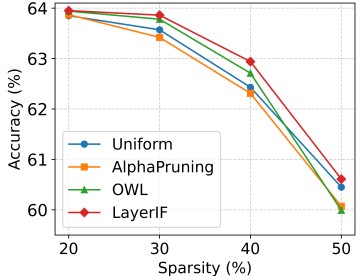

Figure 3: Mean accuracy across 4 sparsity levels (20%–50%) for Mistral-7b-v0.1 pruned using SparseGPT.

accuracy across all pruning ratios, while baseline methods exhibited greater performance variability, frequently changing ranks across sparsity levels. On average, LayerIF outperformed the next best method, OWL, by a 0.37% percentage increase. Moreover, we provide additional layer-wise sparsity allocation results for Gemma-7b in Appendix E and results with and without the smoothing of IF values in Appendix F.

## 5 Discussion

In this section, we provide a discussion of the components of the LayerIF pipeline and elaborate on key design choices and ablation studies conducted as part of our experiments.

**Layer-wise Allocation Differences.** We show a comparison between the number of experts allocated in LayerIF versus other baselines in Figure 4. Unlike AlphaLora and MoLA, which apply the same allocations to every test task, LayerIF dynamically adapts the allocations based on data-driven

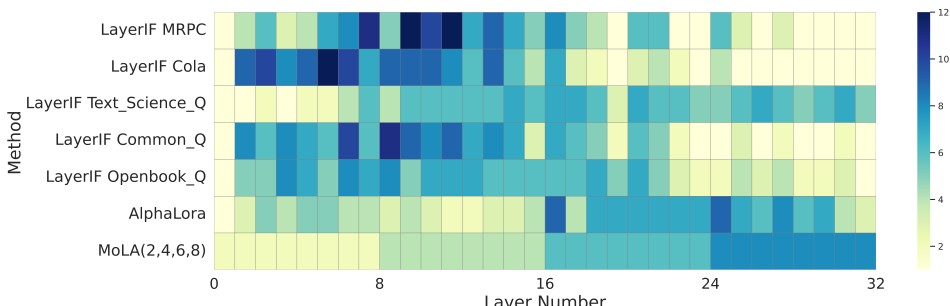

Figure 4: Heatmap showing expert allocations across Transformer layers for Mistral-7B-v0.1 at a total of 160 experts, comparing our dataset-specific approach LAYERIF to ALPHALORA and MOLA. Unlike ALPHALORA and MOLA, which apply the same allocations to every test task, LAYERIF adapts allocations based on the dataset. This is reflected in the diverse allocation patterns across the various LAYERIF rows as darker shades indicate higher expert allocation to that particular layer.

Table 3: Comparison of accuracies across different pruning strategies (Magnitude, Wanda, SparseGPT) between LAYERIF and reversed allocation at 70% pruning on Mistral-7b-v0.1.

| Method | Magnitude | Wanda | SparseGPT | Average |
|---|---|---|---|---|
| LAYERIF | 33.44 | 32.50 | 41.14 | **35.69** |
| Reversed Allocation | 33.10 | 32.02 | 38.45 | 34.52 |

task-affinity estimates. This clearly indicates the unique benefit of our approach, as we can tune the allocation based on how much task affinity a model has, as opposed to prior work, which offers allocation for all datasets based on model-only information at best.

**Interpretation of Layer Sensitivity.** Influence Functions quantify how small perturbations in training data affect the model's loss. In this view, layers that are well-trained and have effectively converged tend to exhibit lower sensitivity to such perturbations, as reflected by smaller IF magnitudes. To further examine this relationship, we conduct an ablation in which we invert our allocation strategy, assigning higher pruning ratios to layers with lower IF sensitivity. Under a 70% sparsity budget on MISTRAL-7B-V0.1, this reversed allocation leads to consistently worse downstream performance, reinforcing that layers with lower IF sensitivity are better trained and should be preserved more aggressively, as can be seen in Table 3.

**Guidelines for Influence Sample Selection.** We provide a empirical perspective on how to select the subset of influence samples used to estimate layer quality. Throughout the paper, we aggregate only positive-influence samples (+VE) by default, as they reflect training points that reduce validation loss and thus carry constructive learning signal. In Section 4, we additionally examine two variants: (ALL) which includes all samples regardless of sign, and (TOP-25%) which retains the top quartile of positive scores by magnitude. Empirically, the TOP-25% variant achieves the highest average performance across datasets, suggesting it is a strong default in practice.

From a theoretical standpoint, this selection serves as a noise-filtering mechanism. Because IF estimates are computed from stochastic Hessian and gradient approximations, small-magnitude scores often correspond to spurious or weak training–validation interactions. Retaining only the upper tail of positive influences increases the signal-to-noise ratio. Prior studies have also shown that IF values in deep models are heavy-tailed [34], implying that a small fraction of training samples account for most of the total positive influence. Hence, focusing on this high-influence subset isolates the most informative training signals.

In practice, one can determine the cutoff ratio using a simple cumulative influence curve: sorting positive IF values in descending order and identifying the elbow point where marginal contribution drops sharply provides a data-driven threshold. This adaptive thresholding remains an exciting direction for future work.

**Per Matrix versus Per Block.** We compute the IF scores on a per-block basis within each Transformer layer. That is to say, the IF scores are aggregated across the attention and MLP heads

of a given layer, and the average score is uniformly applied to all associated weights, rather than computing separate scores for each component. This design choice follows the precedent set by prior work [19, 20]. To empirically assess the impact of this decision, we conducted an ablation study using LLaMA2-7B [11] at 70% sparsity across all three pruning methods. We observed that computing scores on a per-block basis yielded a 2.14% average improvement in accuracy, thereby corroborating the effectiveness of this approach.

**Computational Complexity.** One limitation of IF-based methods in general (at test-time) is their high computational complexity, primarily due to the Hessian inversion. However, optimizing this compute time is an active area of research [29, 82] and our LAYERIF framework can employ any efficient IF method proposed in the future. Additionally, it is important to note that our framework is not for use during inference. That is, since our framework is applied *during model training*, the computational overhead of IF is a minuscule fraction of the time it takes to pretrain LLMs. Detailed wall-clock times and memory consumption for varying model and dataset sizes are presented in Appendix H.

**Comparison with Alternative IF Methods.** One class of alternative IF methods are Hessian-free methods like TracIn [28]. While Hessian-free IF methods are computationally more efficient, they generally lead to a drop in performance [38, 29]. To assess if Hessian-free methods such as TracIn [28] can be used for layer quality influence estimation, we compute the layer-wise expert allocations for each dataset from our main experiments using TracIn as the IF method in LAYERIF and compare these allocations with those derived from the Hessian-based method, DataInf [81] (employed in our main experiments). We compute the Spearman's correlation coefficient and present our findings in Table 4. We find that Hessian-free methods fail to capture meaningful information about layer-wise training quality as there is little to no agreement in the resulting layer allocations across

Table 4: Measuring Spearman's correlation coefficient between Hessian-based Hessian-free methods when employed in LAYERIF.

| Dataset | Correlation |
|---|---|
| MRPC | -0.22 |
| Cola | 0.44 |
| Openbook Q | 0.09 |
| Text Science Q | -0.69 |
| Common Q | 0.12 |

datasets when employing TracIn compared to DataInf in LAYERIF. Another class of alternative IF methods are Gauss-Newton Hessians like the EK-FAC method [34]. The main bottleneck with the Gauss-Newton Hessian is the time required for IF computation. In our experiments, the Gauss Newton Hessian was roughly 160× slower than using the DataInf approximation, making DataInf the only practical choice for our large-scale experiments.

## 6 Conclusion

In this work, we introduced LAYERIF, a novel framework that leverages Influence Functions (IFs) to quantify layer quality in large language models (LLMs). Departing from prior model-centric information and heuristics, our approach provides a data-informed perspective by capturing the sensitivity of model loss to individual training examples at the layer level. To validate the utility of our layer-wise IF estimates, we applied them to two practical downstream tasks: expert allocation in Mixture-of-Experts (MoE) architectures and layer sparsity allocation for model pruning. Our method consistently outperformed the previously best approaches, achieving a 1.61% improvement in LoRA-MoE and a 0.90% gain in zero-shot accuracy after pruning. These results confirm that IF-based layer quality estimation can guide more effective structural adaptations in LLMs. Beyond its empirical gains, LAYERIF offers a general, training-data-aware framework for interpreting and optimizing deep models, applicable across architectures and deployment scenarios.

## Acknowledgment

We appreciate the reviewers for their insightful comments and suggestions, and for helping strengthen our work. Hadi Askari and Muhao Chen were supported by the NSF of the United States Grants ITE 2333736, OAC 2531126, and the Amazon Nova Trusted AI Prize. Anshuman Chhabra was supported by the USF Faculty Startup Fund.

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

# Appendix

## A Limitations

A key limitation of Influence Functions is their reliance on an implicit convexity assumption, which may not strictly hold in LLMs. While this is a known limitation of IF theory and an ongoing area of research, we find that our proposed LAYERIF framework demonstrates strong empirical performance on LLMs despite this. We also note that LAYERIF is a training-time method and does not impact inference latency. Although computing IFs can be computationally intensive, particularly due to the Hessian inverse approximation, recent approaches such as DataInf offer closed-form solutions that significantly reduce this overhead. Future work will explore integrating such scalable IF estimators to further improve the efficiency of our method. Another limitation of our method stems from its data-driven nature: the subset of data used to compute IF scores must be representative of the overall training and validation distribution. Failure to ensure representativeness can lead to biased or unreliable influence estimates, potentially degrading the effectiveness of the method.

## B Broader Impact

Our data-driven approach to assessing layer quality using Influence Functions offers a promising direction for making pretrained LLMs more efficient for downstream tasks. By enabling more targeted pruning, models can be made both more lightweight and environmentally sustainable. Additionally, optimizing expert allocation can enhance the performance of smaller models on specialized tasks, further improving efficiency and reducing computational and environmental costs. Our method also contributes to model interpretability by identifying which layers are most responsible for performance on specific downstream tasks. This insight can guide further research, ultimately leading to more effective and efficient language models.

## C Clarifying Walk-Through Example

Here, we provide a concrete step-by-step walk-through of our method for greater clarity of our method.

**Concrete walk-through for expert allocation (CommonQ dataset, 32-layer LLM, T=160 experts, $\beta = 3$)**

Below we provide the exact numbers produced by our implementation. For brevity only the first 5 layers are shown; the remaining layers follow identically.

| Layer $i$ | Raw IF score $v_i$ | Inverted $\tilde{v}_i$ | Power-scaled $\hat{v}_i$ | Fractional $f_i$ | Final experts $s_i$ |
|---|---|---|---|---|---|
| 0 | $-5.01 \times 10^{11}$ | $5.01 \times 10^{11}$ | $1.26 \times 10^{35}$ | 1.12 | **2** |
| 1 | $-8.36 \times 10^{11}$ | $8.36 \times 10^{11}$ | $5.84 \times 10^{35}$ | 5.21 | **6** |
| 2 | $-7.99 \times 10^{11}$ | $7.99 \times 10^{11}$ | $5.10 \times 10^{35}$ | 4.55 | **6** |
| 3 | $-8.46 \times 10^{11}$ | $8.46 \times 10^{11}$ | $6.05 \times 10^{35}$ | 5.40 | **6** |
| 4 | $-8.16 \times 10^{11}$ | $8.16 \times 10^{11}$ | $5.43 \times 10^{35}$ | 4.84 | **6** |
| . . . | . . . | . . . | . . . | . . . | . . . |

- **Step1 (sign inversion).** Invert the negative values to positive.
- **Step2 (power transform).** Amplify disparities by raising to the power $\beta$ to get $\hat{v}$.
- **Step3 (normalise to budget).** Scale $\hat{v}$ so that the fractional allocations $\sum f_i$ equal $\mathbf{T} - \mathbf{m} = \mathbf{128}$ extra experts ($m = 32$ layers).
- **Step4 (floor + redistribute).** Set $s_i = \lfloor f_i \rfloor + 1$, guaranteeing $\geq 1$ expert/layer, then distribute the remaining 24 units to the layers with the largest fractional parts.

Finally, the complete allocation for **CommonQ** becomes:

$$[2, 6, 6, 6, 6, 5, 7, 6, 8, 7, 6, 7, 6, 6, 6, 5, 6, 6, 5, 4, 5, 5, 4, 2, 4, 5, 3, 4, 2, 3, 4, 3]$$

**Interpretation.** The middle layers receive the most experts, indicating they are comparatively under-trained for CommonQ; the first and very deep layers receive lower number of experts, reflecting higher estimated quality.

## D  Gemma-7b Expert Allocation Results

This section presents results on layer-wise expert allocation for Gemma-7B, using a total of 160 experts in Table 5.

Table 5: Accuracy across datasets using ALPHALORA, MOLA(3,5,7,8) and LAYERIF (+VE) for 160 experts on Gemma-7b.

| Dataset | ALPHALORA | MOLA(3,5,7,8) | LAYERIF (+VE) |
|---|---|---|---|
| MRPC | 82.61 | 82.89 | 84.75 |
| Cola | 86.09 | 86.86 | 86.96 |
| Text Science Q | 94.15 | 93.88 | 93.66 |
| Common Q | 84.11 | 83.94 | 83.46 |
| Openbook Q | 87.80 | 88.80 | 88.20 |
| Average | 86.55 | 87.47 | **87.81** |

As shown in the results, LAYERIF achieves superior average performance compared to the baseline methods.

## E  Gemma-7b Sparsity Allocation Results

Here, we provide results on layer-wise sparsity allocation for Gemma-7B for 50% sparsity in Table 6.

Table 6: Comparison of average accuracies across different pruning strategies (Magnitude, Wanda, SparseGPT) and layer-wise sparsity allocation methods at 50% pruning on Gemma-7b.

| Method | Magnitude | Wanda | SparseGPT | Average |
|---|---|---|---|---|
| UNIFORM | 32.21 | 51.19 | 49.92 | 44.44 |
| ALPHAPRUNING | 34.21 | 50.76 | 51.39 | 45.45 |
| OWL | 33.27 | 47.15 | 47.48 | 42.63 |
| LAYERIF | 32.94 | 53.01 | 50.90 | **45.62** |

We can again see that LAYERIF has both the highest average and the best individual performance after pruning.

## F  Sparsity Allocation Results with and without smoothing for Mistral-7b-v0.1

As described in Section 3, we apply Savitzky–Golay smoothing [91] to the min-max normalized IF values within our layer-wise sparsity allocation algorithm. This smoothing step helps prevent any individual layer from being entirely pruned, while also enhancing the robustness and consistency of sparsity assignments. This was particularly helpful in scenarios where raw influence scores fluctuate significantly between adjacent layers.

We used a window-length of 7 and a polyorder of 3 for our filter. We present the percentage change[3] in performance at 70% sparsity, comparing results with and without smoothing, for Mistral-7B-v0.1 in Table 7.

We can clearly see that smoothing leads to an overall increase in performance across the different epsilon values and pruning methods.

---

[3]Percentage change is computed as $\left( \frac{\text{with smoothing} - \text{without smoothing}}{\text{without smoothing}} \right) \times 100\%$

Table 7: Percentage change between zero-shot accuracies across different $\epsilon$ values and LLM pruning methods.

|  | **Magnitude** | **Wanda** | **SparseGPT** |
|---|---|---|---|
| $\epsilon = 0.1$ | (+) 1.49 | (+) 0.94 | (+) 0.97 |
| $\epsilon = 0.2$ | (+) 0.52 | (+) 1.15 | (+) 0.46 |
| $\epsilon = 0.3$ | (+) 0.03 | (+) 0.05 | (+) 0.41 |
| $\epsilon = 0.4$ | (+) 1.81 | (-) 0.69 | (+) 0.48 |
| $\epsilon = 0.5$ | (+) 0.63 | (+) 0.78 | (-) 0.02 |

## G    Statistical Significance of Results

We present the t-test statistical significance of our empirical results across both subsettings. Specifically, we analyze the average accuracy of LAYERIF compared to baseline methods across all expert counts and datasets evaluated for layer-wise expert allocation, as shown in Table 8.

Table 8: Computing the t-test metric to compare LAYERIF with MOLA and ALPHALORA on zero-shot accuracy across the different number of experts (** indicates statistical significance i.e., $p$-value $< 0.05$)

| Algorithm | Average Accuracy | $p$-value |
|---|---|---|
| ALPHALORA | 79.30 | 0.03152** |
| MOLA | 78.97 | 0.03123** |
| LAYERIF | 82.28 | - |

Table 8 clearly shows a statistically significant improvement over both of the other baselines.

Table 9: Computing the t-test metric to compare LAYERIF with ALPHAPRUNING, UNIFORM and OWL on zero-shot accuracy across the different number of pruning ratios (** indicates statistical significance i.e., $p$-value $< 0.05$)

| Algorithm | Average Accuracy | $p$-value |
|---|---|---|
| ALPHAPRUNING | 62.42 | 0.01969** |
| UNIFORM | 62.58 | 0.03077** |
| OWL | 62.61 | 0.09156 |
| LAYERIF | 62.84 | - |

We now analyze the average accuracy of LAYERIF compared to baseline methods across all pruning ratios for layer-wise sparsity allocation, as shown in Table 9.

Table 9 clearly shows that LAYERIF produces statistically significant improvements in all of the baselines, except OWL, where it shows a marginally significant improvement.

## H    Wall Clock Times and Memory Consumption for Varying Dataset and Model Sizes

We report the wall-clock times of our Influence Function (IF) computation using the DATAINF approximation, compared against the TRACIN baseline, under varying dataset sizes and model scales. All experiments were performed on $4\times$ NVIDIA RTX 6000 Ada Generation GPUs.

### H.1    Scalability to Dataset Size

Table 10 presents runtimes for computing IFs over a single layer with increasing numbers of training and validation samples. As expected, the runtime scales approximately linearly with dataset size. Although DATAINF is substantially slower than TRACIN, the total cost remains modest relative to the computational expense of pretraining large language models. For comparison, training LLAMA2-70B reportedly required $\sim$1.72M GPU hours, whereas DATAINF adds only a minor diagnostic overhead that can be amortized across layers to identify underperforming components during training.

The peak GPU memory consumption during influence computation for a dataset of 300 training and 50 validation samples was approximately 35000MiB for the 7B model. This footprint remains

Table 10: Wall-clock runtimes (in seconds) for computing IFs for a single layer under different dataset sizes on MISTRAL-7B-V0.1.

| Dataset Size (Train×Val) | Method | Runtime (s) |
|---|---|---|
| 3000×500 | DATAINF | 163,721 |
| | TRACIN | $1.35 \times 10^{-2}$ |
| 300×50 | DATAINF | 402.77 |
| | TRACIN | $3.31 \times 10^{-5}$ |
| 30×5 | DATAINF | 15.3 |
| | TRACIN | $1.19 \times 10^{-6}$ |

well within the typical per-GPU memory budget, indicating that influence-based diagnostics can be feasibly integrated into existing training pipelines.

## H.2 Scalability to Model Size

We also evaluate scaling across model sizes in Table 11. The runtime grows roughly proportionally to parameter count.

Table 11: Wall-clock runtimes (in seconds) for DATAINF and TRACIN under different model sizes (300×50 dataset).

| Model | Method | Runtime (s) |
|---|---|---|
| QWEN2.5-3B | DATAINF | 58.063 |
| | TRACIN | $2.79 \times 10^{-5}$ |
| QWEN2.5-32B | DATAINF | 2,824.16 |
| | TRACIN | $6.82 \times 10^{-5}$ |

Overall, while DATAINF incurs higher per-layer computation cost than first-order approximations such as TRACIN, it remains computationally tractable even for multi-billion-parameter models. Its cost is negligible relative to the full pretraining budget, while providing rich layer-wise diagnostic signals that can inform targeted retraining or pruning strategies.

# I Code and Reproducibility

We follow the datasets and evaluation protocol of AlphaLora [47] for layer-wise expert allocation and the datasets and evaluation protocol of Alphapruning [19] for layer-wise sparsity allocation. Our code has been anonymized and uploaded here: `https://github.com/HadiAskari/Expert_Allocation` and `https://github.com/HadiAskari/LayerIF_Pruning_New`.

