# OpenReview forum: "LayerIF: Estimating Layer Quality for Large Language Models using Influence Functions"
_NeurIPS.cc/2025/Conference — NeurIPS 2025 poster_

### Official Review · Reviewer_zyf6 · 2025-06-19

**Clarity:** 2
**Significance:** 2
**Originality:** 2
**Rating:** 4
**Confidence:** 3

**Summary:**

This paper proposes LayerIF, an influence function-based method for estimating the importance of individual layers across various downstream tasks. The method is evaluated through experiments on expert allocation in LoRA-MoE architectures and layer-wise sparsity distribution for LLM pruning, demonstrating its effectiveness.

**Questions:**

Please see Strengths And Weaknesses

**Ethical Concerns:**

["NO or VERY MINOR ethics concerns only"]

**Final Justification:**

The authors have addressed some of my concerns during the rebuttal period. Although certain limitations remain — such as limited generality and seemingly marginal improvements — I recommend an overall score of 4.

**Limitations:**

yes

**Paper Formatting Concerns:**

N.A.

**Quality:**

3

**Strengths And Weaknesses:**

Strengths:
1.    The paper is well written, with clear motivations and a thorough explanation of the proposed methodology.

2.    The connections to prior work are appropriately discussed, situating LayerIF within the context of existing research.

Weaknesses and Questions:

1.    In lines 32–33, the classification of activation-based methods as "model-only heuristics or information" may not be entirely accurate. These methods typically require training data to obtain activations for estimating layer importance. Therefore, shouldn't they also be considered at least partially data-centric, similar to LayerIF?

2.    In Figure 1, the results lack context. What specific experiment or task do these results correspond to? What baseline are they compared against? Since the figure includes concrete values, instead of using some likely imaginary numbers, the authors should provide details about the experimental setup that generated these results.

3.    The authors mention in lines 191–192 that “layers with lower cumulative positive influence exhibit less sensitivity to training data, indicating greater stability and training maturity.” This is an intuitive claim. Can the authors conduct a reverse experiment (e.g., prioritizing layers with higher influence scores) to empirically validate this hypothesis?

4.    Another concern is the algorithmic inconsistency between expert allocation and pruning. Specifically, both applications use LayerIF only as an initial step, followed by distinct procedures for score calculation, normalization, or finalization. This suggests that LayerIF may not yet be a well-established or truly plug-and-play method for downstream tasks. Instead, different applications may require varying levels of task-specific post-processing to make effective use of LayerIF.

5.    In lines 296–297, how are the 1.61% and 3.54% performance metrics calculated? Please clarify the source and computation of these figures.

6.    Similarly, in lines 311–312, how are the reported improvements of 1.03% and 0.9% obtained? I couldn't obtain the same numbers.

7.    The improvement from LayerIF in the model pruning experiments appears marginal. Compared to the uniform sparsity ratio, all reported gains are less than 0.5%, which raises concerns about practical significance.

---

> ### Author Rebuttal · Authors · 2025-07-31
>
> We are grateful to the reviewer for their detailed feedback of our work, and are appreciative of their time, efforts, and consideration. Below we provide more discussion on the points raised.
>
> * **Partially data-driven analysis methods:**
>
>     We thank the reviewer for raising this important clarification. We agree that activation-based methods such as those proposed in [20–23] do typically require access to training (or validation) data to compute layer-wise activations, and therefore are not entirely “model-only” in the strictest sense.
>
>     Our original phrasing in Lines 32–33 aimed to distinguish **methods that rely primarily on internal model representations** (e.g., weight norms, activations, attention maps) from those that **explicitly quantify model–data interactions**, such as Influence Functions. While activation-based techniques indeed use input data to probe activations, they do not trace the *causal effect* of individual training examples on model predictions by analyzing the loss. In contrast, Influence Functions estimate the  impact on the loss of perturbing or removing training samples.
>
>     We will revise our language in the introduction to clarify this distinction and cite other papers that are more absolutely model-centric. A more accurate phrasing would be:
>
>     > “Many approaches take a model-centric perspective, examining how the model’s learned weights effect predictions [1,2]. Other methods use activations, or attention patterns to deduce internal mechanisms [3,4,5,6]. In contrast, Influence Functions offer a more explicitly data-centric perspective by quantifying the effect of individual training points on model's loss landscape.”
>
>     We appreciate the reviewer’s suggestion and will make this clarification in the final version.
>
> ___
> ___
>
> * **Clarifying Figure 1:**
>
>     Figure 1 is just a hypothetical illustration (not with real numbers) of our method against model only or heuristic based methods. We will add this clarification in the caption.
>
>     To improve the clarity of our method we will aim to provide a concrete step-by-step walk-through, in the revised version of the paper, as described below:
>
>     **Concrete walk‑through for expert allocation (CommonQ dataset, 32‑layer LLM, T=160 experts, beta=3)**
>
>     Below we provide the exact numbers produced by our implementation.
>     For brevity only the first 5 layers are shown; the remaining layers follow identically.
>
>     | Layer $i$ | Raw IF score $v_i$ | Inverted $ṽ_i$ | Power‑scaled $v̂_i$ | Fractional $f_i$ | Final experts $s_i$ |
>     | --------- | --------------------------------------- | ------------------------------------------------------------- | -------------------------------------------------------------- | ------------------------ | --------------------------- |
>     | 0         | -5.01×10¹¹                            | 5.01×10¹¹                                                        | 1.26×10³⁵                                                              | 	1.12                    | **2**                       |
>     | 1         | -8.36×10¹¹                            | 8.36×10¹¹                                                    | 5.84×10³⁵                                                    | 5.21                    | **6**                       |
>     | 2         | -7.99×10¹¹                            | 7.99×10¹¹                                                   | 5.10×10³⁵                                                    | 4.55                     | **6**                       |
>     | 3         | -8.46×10¹¹                            | 8.46×10¹¹                                                    | 6.05×10³⁵                                                    | 5.40                     | **6**                       |
>     | 4         | -8.16×10¹¹                            | 8.16×10¹¹                                                   | 5.43×10³⁵                                                    | 4.84                     | **6**                       |
>     | …         | …                                       | …                                                             | …                                                              | …                        | …                           |
>
>
>
>   (1) **Step1 (sign inversion).** Invert the negative values to positive.
>
>   (2) **Step2 (power transform).** Amplify disparities by raising to the power beta to get v̂.
>
>   (3) **Step3 (normalise to budget).** Scale v̂ so that the fractional allocations ∑fi equal **T−m=128** extra experts (m=32 layers).
>
>   (4) **Step4 (floor+redistribute).** Set si=⌊fi⌋+1, guaranteeing ≥1 expert/layer, then distribute the remaining 24 units to the layers with the largest fractional parts.
>
>   Finally, the complete allocation for **CommonQ** becomes:
>
>     ```[2, 6, 6, 6, 6, 5, 7, 6, 8, 7, 6, 7, 6, 6, 6, 5, 6, 6, 5, 4, 5, 5, 4, 2, 4, 5, 3, 4, 2, 3, 4, 3]```
>
>     **Interpretation.** The middle layers receive the most experts, indicating they are comparatively under‑trained for CommonQ; the first and very deep layers receive lower number of experts, reflecting higher estimated quality.
>
>     Similarly we will also provide an example for the layer-wise sparsity allocation setting. If the reviewer would like us to take any other steps to improve clarity, we are more than happy to do so.
>
> ___
> ___
>
> * **Reverse experiment**:
>
>     We thank the reviewer for this suggestion. We provide an ablation where we reversed our methodology by pruning the less sensitive layers more at a 70\% sparsity budget on Mistral-7b-v0.1 and find that our design choice leads to improved performance over reversed allocation. We present the results below.
>
>
>
>     |             Method  | Magnitude | Wanda  | SparseGPT | Average  |
>     |---------------------|-----------|--------|-----------|----------|
>     | Our Method          | **33.439**    | **32.497** | **41.143**    | **35.693**   |
>     | Reversed Allocation | 33.101    | 32.015 | 38.450    | 34.522   |
>
>
>     We will add this ablation to the revised version of our paper.
>
> ___
> ___
>
> * **LayerIF having algorithmic inconsistency:**
>
>     We would like to clarify the generality of our approach. Influence Functions (IFs), by design, are diagnostic tools intended to inform model developers about the sensitivity of model components to training data. LayerIF extends this diagnostic capability to estimate per-layer quality in a principled, data-driven manner.
>
>     While it is true that expert allocation and pruning use distinct post-processing steps after the initial LayerIF score computation, this is by design: each downstream application naturally requires a different mapping from layer quality estimates to actionable decisions (e.g., allocating experts vs. assigning sparsity levels). However, the core LayerIF signal remains the same and is easily integrated across use case.
>
>      Importantly, LayerIF is not specific to the two tasks studied in this paper. It is a general-purpose tool that can be applied in the future to any task where layer quality matters (e.g. layer-wise unlearning). Its ability to improve performance across diverse tasks and compete against methods that are tailor to those specific tasks, in our view, is a strength rather than a limitation.
>
> ___
> ___
>
> * **Clarification on numbers:**
>
>     All these numbers follow the relative change formula of (New-Old)/Old * 100. We provide the calculations below:
>
>     * **1.61:** IF (`Top 25%`) and MoLA-8642. (82.27-80.97)/80.97 * 100.
>     * **3.54:** Average of IF variants and Average of baselines. (81.96-79.175)/79.175 * 100. This is a typo the correct value is 3.52. We will change this.
>     * **1.03:** IF SparseGPT and OWL SparseGPT. (60.61-59.99)/59.99 * 100.
>     * **0.90:** IF SparseGPT and AlphaPruning SparseGPT. (60.61-60.07)/60.07 * 100.
>
>     We hope this clears the confusion on how we arrived on these numbers.
>
> ___
> ___
>
> * **Improvement appears marginal:**
>
>     We appreciate the reviewer’s concern and would like to offer some clarification. While the absolute gains over the uniform sparsity baseline in the pruning experiments may appear numerically modest (under 0.5\%), this level of improvement is consistent with recent state-of-the-art work that we compare against OWL (ICML 2024) and AlphaPruning (NeurIPS 2024).
>
>     Moreover, our method is not designed exclusively for pruning. Model pruning was included as one illustrative downstream task to demonstrate the applicability of LayerIF, not as the core objective of the paper. LayerIF is a general-purpose diagnostic tool, and its effectiveness across multiple settings speaks to its broader utility beyond the pruning context.
>
>     Thus, we believe that providing a unified, data-driven method that performs competitively across such diverse applications is a notable contribution.
>
> ---
> ___
>
> We thank the reviewer for their comments and we hope that we provided sufficient clarifications. We hope to discuss further in the discussion period in case any others remain. Thank you.
>
> ___
> ---
>
> References:
>
> [1]. Yin, Lu, et al. "Outlier weighed layerwise sparsity (owl): A missing secret sauce for pruning llms to high sparsity." arXiv preprint arXiv:2310.05175 (2023).
>
> [2]. Lu, Haiquan, et al. "Alphapruning: Using heavy-tailed self regularization theory for improved layer-wise pruning of large language models." Advances in neural information processing systems 37 (2024): 9117-9152.
>
> [3]. Serrano, Sofia, and Noah A. Smith. "Is attention interpretable?." arXiv preprint arXiv:1906.03731 (2019).
>
> [4]. Abnar, Samira, and Willem Zuidema. "Quantifying attention flow in transformers." arXiv preprint arXiv:2005.00928 (2020).
>
> [5]. Wiegreffe, Sarah, and Yuval Pinter. "Attention is not not explanation." arXiv preprint arXiv:1908.04626 (2019).
>
> [6]. Hao, Yaru, et al. "Self-attention attribution: Interpreting information interactions inside transformer." Proceedings of the AAAI Conference on Artificial Intelligence. Vol. 35. No. 14. 2021.

---

> > ### Comment · Reviewer_zyf6 · 2025-08-01
> >
> > Thank you to the authors for the clarifications. Some of my concerns have been addressed; however, the following three points remain:
> >
> > Regarding Figure 1, it would be better to plot results using real experimental data instead of hypothetical (imaginary) numbers, especially since you also compare against baselines in this figure. It is not fair to use non-existent numbers to criticize baseline methods quantitatively. Qualitative comparisons based on hypotheses are acceptable, but you chose to make a quantitative comparison using non-existent data, which is both unfair and inconsistent with the rigor expected in academic papers.
> >
> > It is unusual that the authors chose to present results in terms of relative improvement rather than absolute improvement. At least in the pruning field, all prior work reports absolute improvements. Using relative values gives the impression that the performance gains are being exaggerated.
> >
> > Regarding the marginal improvements, I find that your results do not align with the performance reported in the original OWL paper. I have read the OWL paper, and it reports a 2%–4% improvement for both Wanda and SparseGPT. However, in your paper, OWL even performs worse in most cases. Could this be due to differences in the backbone models used (e.g., they used LLaMA, whereas you used Mistral)? Therefore, the explanation provided for the marginal improvements is not convincing to me at this stage.

---

> ### Author Response · Authors · 2025-08-02
> **Reply to Response**
>
> The authors thank the reviewer for engaging thoughtfully with our rebuttal. Below, we address the specific points raised:
>
> * **Figure 1:** We acknowledge the reviewer’s concerns and will remove all hypothetical numbers from the figure in the revised version. This change is straightforward to implement.
>
> * **Relative Improvement:** We apologize for the confusion. Our use of relative improvement metrics was guided by conventions in the area, where such reporting is common—for example, see Section 4.2 in [1] and Figure 30 in [2]. That said, we are happy to include the corresponding absolute values in the text to improve clarity.
>
> * **Comparison to OWL:** We also observed that OWL underperforms relative to the baselines in certain settings; however, we note that OWL was the second-best performing configuration at three out of the four sparsity levels reported in Figure 3. This trend is even more pronounced in our Gemma‑7B results (included in the supplementary and reproduced below). We attribute these differences to variations in model architecture, though a deeper investigation is warranted and could be pursued in future work.
>
>
> | Method             | Magnitude | Wanda | SparseGPT | Average |
> |--------------------|-----------|-------|------------|---------|
> | **Uniform**        | 32.21     | 51.19 | 49.92      | 44.44   |
> | **AlphaPruning**   | 34.21     | 50.76 | 51.39      | 45.45   |
> | **OWL**            | 33.27     | 47.15 | 47.48      | 42.63   |
> | **LayerIF**    | 32.94     | 53.01 | 50.90      | **45.62** |
>
> ---
>
> We thank the reviewer for engaging, hopefully our further clarifications warrant an increase in score.
>
> ---
>
> References:
>
> [1]. Abhineet Agarwal, Yan Shuo Tan, Omer Ronen, Chandan Singh, Bin Yu Proceedings of the 39th International Conference on Machine Learning, PMLR 162:111-135, 2022.
>
> [2]. Grosse, Roger, et al. "Studying large language model generalization with influence functions." arXiv preprint arXiv:2308.03296 (2023).

---

> ### Comment · Reviewer_zyf6 · 2025-08-05
>
> Thank the authors for the follow-up responses. Some of my concerns are addressed. Just on point: in your response, the improvement compared to AlphaPruning seems margnial. However, based on your your detailed and prompt reponses in the whole rebuttal period, I will increase my scores accordingly.

---

> > ### Comment · Reviewer_zyf6 · 2025-08-07
> > **update**
> >
> > After careful consideration over the past few days, I’ve decided to raise my score to 4.

---

> > > ### Author Response · Authors · 2025-08-09
> > > **Reply to Response**
> > >
> > > We thank the reviewer for their engagement and changing their score from a 2 to a 4. We are grateful for the additional experiments that helped us improve our paper.

---

### Official Review · Reviewer_df7d · 2025-07-01

**Clarity:** 2
**Significance:** 3
**Originality:** 3
**Rating:** 4
**Confidence:** 4

**Summary:**

This paper introduces LayerIF, a novel framework for estimating the layer-wise quality in Large Language Models (LLMs) using Influence Functions (IFs). LayerIF quantifies layer quality by measuring each layer’s sensitivity to training data, producing task-specific importance scores. The authors validate their approach through two downstream applications: (1) data-driven expert allocation in Mixture-of-Experts (MoE) architectures, which optimises the distribution of LoRA adapters per layer , and (2) layer-wise sparsity distribution for LLM pruning, where layers identified as being of lower quality are pruned more aggressively.

**Questions:**

1. Please explain the correct form of the influence function, $I(z_i)$, and the layer quality score, $S^{(l)}$, in the paper.
2. Could a theoretical analysis or an empirical ablation study be provided to support the formulation of $S^{(l)}$, which involves summing only the positive influences?
3. In the MoE expert allocation section, please explain the relationship between $v_i$ and $ S^{(l)}$. If $v_i = S^{(l)}$, please clarify the definition for the function $\tilde{v_i}^{\beta}$.
4. Please explain the computational efficiency, approximation error, and memory overhead associated with the calculation of the Hessian inverse.

**Ethical Concerns:**

["NO or VERY MINOR ethics concerns only"]

**Final Justification:**

The reviewer’s concerns have been addressed. The reviewer appreciates the authors’ efforts to improve the work and has raised the score to 4.

**Limitations:**

Yes.

**Paper Formatting Concerns:**

No.

**Quality:**

2

**Strengths And Weaknesses:**

Strengths
1. The work addresses a critical gap in LLM efficiency: data-driven layer optimisation. The framework's ability to inform task-specific structural adaptations, such as dynamic expert allocation, is a significant step towards the more efficient deployment of LLMs.
2. To my knowledge, this paper is the first to leverage Influence Functions for the purpose of estimating layer quality in LLMs. This presents a novel application of a classic statistical technique and provides a new, data-centric lens for analysing model internals.

Weaknesses
1. Inconsistency in Formulation: The paper's formulation of the Influence Function contains a significant theoretical inconsistency that creates considerable confusion. The definition of
$I(z_i)$ on page 4 is missing a leading negative sign when compared to the original formulation in the cited reference [26] (Understanding Black-box Predictions via Influence Functions). While the mathematical formula is different, the paper's textual interpretation—that a positive influence score corresponds to a detrimental sample that increases the loss —aligns with the original work. This discrepancy between the paper's mathematical definition and its textual interpretation is a notable flaw.
2. Lack of Justification for Score Aggregation: The paper defines the layer quality score, $S^{(l)}$, by exclusively summing the positive influences. It is unclear why negative influences are excluded. The rationale for this choice over alternatives, such as  the net sum of positive and negative influences, is not provided. Theoretical support or ablation study comparing these different aggregation strategies would be necessary to validate this design.
3. Lack of Clarity in MoE Allocation Method: Specifically, it introduces a new variable, $v_i$, to represent the layer score without explicitly stating its relationship to the previously defined score $S^{(l)}$ from Section 3.2. If $v_i$ is equal to $S^{(l)}$ and $S^{(l)}$ is a positive number, then $\tilde{v_i}$ is negative. Then For any non-integer value of $\beta>0$, $\tilde{v_i}^{\beta}$ is not well-defined within the real numbers and would yield complex values.
4. Insufficient Analysis of Computational Feasibility: The paper claims computational efficiency can be achieved by leveraging recent methods like DataInf, but it does not provide a concrete analysis of this. A quantitative comparison of the runtime, approximation error and memory overhead is needed to substantiate claims of practical efficiency.
5. Minor Issues
(1) Line 89, "pruninng" should be "pruning".
(2) There is an inconsistency in the model version used. The text states experiments were conducted on Mistral-7b-v0.3 , but the captions for Table 2 and Figure 2 specify Mistral-7b-v0.1.
(3) In Appendix G, "laywer-wise" should be "layer-wise".
(4) In the title for Table 4 in the Appendix, "accross" should be "across".

---

> ### Author Rebuttal · Authors · 2025-07-31
>
> We are grateful to the reviewer for their detailed feedback of our work, and are appreciative of their time, efforts, and consideration. Below we provide more discussion on the points raised.
>
> * **Inconsistency in formulation:**
>
>     Thank you for pointing out this typo, we appreciate it. We sincerely apologize for the missing (leading) '-' sign in the formulation, and will easily make this correction in the paper revision. However, after adding the - sign, the rest of the paper can remain unchanged and no other major changes are required.
>
>     We would also like to mention that the typo in the formulation does not change the validity of our experimental design or the results of our experiments, and we apologize for the inconsistency once again.
>
> ___
> ___
>
> * **Lack of Justification for Score Aggregation:**
>
>     We thank the reviewer for raising this point. Our choice to aggregate only positive influence values was guided primarily by empirical evidence rather than theoretical assumptions. To validate this design choice we conducted this ablation in Section 4.1 where the `Top 25%` variant was the highest performing. Additionally, we perform another ablation in the rebuttal period where we compare keeping all of the samples against keeping only the positive samples on the Gemma-7b model.
>
>
>     | Dataset            | IF (+ve)  | IF (ALL) |
>     |--------------------|------------------|-----|
>     | MRPC               | 84.75            | 82.49 |
>     | CoLA               | 86.96            | 85.14 |
>     | Common_Q           | 83.46            | 82.06 |
>     | Openbook_Q         | 88.20            | 86.20 |
>     | Text_Science_Q     | 93.66            | 94.56 |
>     | **Average**        | **87.81**        | **86.09** |
>
>
>     Here, we see that IF (+ve) was significantly better performing on average and individually more performant on 4 out of the 5 datasets.
>
>     We acknowledge that this does not preclude the possibility of more nuanced aggregation schemes. Future work could explore adaptive thresholds or weighting strategies that combine both helpful and harmful samples in a principled way. We will expand our discussion to acknowledge this broader design space in the revised version of this paper.
>
> ___
> ___
>
> * **Lack of Clarity in MoE Allocation Method:**
>
>     We apologize for the lack of clarity. The $v_i$ is indeed corresponding the the individual layer scores in $S^{(l)}$. Since our implementation followed the DataInf convention as described above, the layerwise values in $S^{(l)}$ were all negative, hence the inversion and then power transformation did not lead to complex values. We will revise this and make things consistent in the paper.
>
> ___
> ___
>
> * **Computational feasibility for DataInf and TracIn under varying datasets and model sizes:**
>
>     We thank the reviewer for raising the important point regarding the computational times of Influence Function methods. Below, we provide wall-clock timings for computing the influence function for a single layer across varying training and validation set sizes. We used 4× NVIDIA RTX 6000 Ada Generation GPUs to perform these experiments.
>
>
>     | Scalability to different Dataset Sizes | Method  | Runtime (seconds) |
>     | -------------------- | ------- | --------------------------- |
>     | Mistral-7B-v0.1, 3000×500         | DataInf | 163,721                     |
>     |                      | TracIn  |  0.0135    |
>     | Mistral-7B-v0.1, 300×50           | DataInf | 402.77                      |
>     |                      | TracIn  | 3.314×10^-5                |
>     | Mistral-7B-v0.1, 30×5             | DataInf | 15.3                        |
>
>
>     As we can see the time increases non-linearly with larger datasets, but we intend to clarify that in our experiments, having approximately 300-500 training samples and 50-100 validation samples is typically sufficient to gauge how well an LLM layer generalizes to a task. Moreover, the current implementation of DataInf runs primarily on CPU; preliminary tests show that offloading key computations to the GPU yields up to a 2× speedup, indicating substantial room for further efficiency improvements.
>
>
>     Similarly, we report the runtime for computing the influence function for 1 layer for different model sizes.
>
>
>     | Scalability to different Model Sizes | Method  | Runtime (seconds) |
>     | -------------------- | ------- | --------------------------- |
>     |                      | TracIn  | 1.192×10^-6                |
>     | Qwen2.5‑3B, 300×50   | DataInf | 58.063                      |
>     |                      | TracIn  | 2.789×10^-5                 |
>     | Qwen2.5‑32B, 300×50  | DataInf | 2,533.46                    |
>     |                      | TracIn  | 5.245×10^-6                 |
>
>
>     As expected, larger models incur higher influence computation times; however, these costs are negligible relative to the overall expense of pretraining large language models. For context, training LLaMA2‑70B required approximately 1.72 million GPU hours. By comparison, our method represents a minor overhead and can be integrated into the broader pretraining pipeline to diagnose underperforming layers and nudge the model toward more uniform training dynamics. This can support developers in identifying and addressing layer‑specific inefficiencies during model development.
>
>
>     Furthermore, the peak GPU memory usage during influence computation for a dataset with 300 training and 50 validation samples was approximately 35,000 MiB for a 7B parameter model. This footprint is relatively modest given the model size and suggests that influence-based diagnostics can be feasibly incorporated into GPU-based training pipelines without exceeding typical memory constraints.
>
>
>     Finally, for error analysis, we defer to the approximation error bounds provided in [1], as we directly employ their DataInf method to compute influence scores. To further validate the reliability of these scores, we compare them against alternative estimators, notably TracIn, and demonstrate its limitations as a proxy for layer quality in our setting (possibly owing to the Hessian being discarded).
>
> ___
> ___
>
> * **Minor Issues:** Thank you for pointing these out, and apologies for missing these prior to submission. We shall fix these small errors and typos.
>
> ---
> ___
>
> Thank you once again for all your efforts in helping to strengthen our work. We hope that our comments helped address your concerns and look forward to engaging with you more during the author-reviewer discussion phase regarding any others that remain.
>
> ___
> ---
>
> References:
>
> [1]. Kwon, Yongchan, et al. "Datainf: Efficiently estimating data influence in lora-tuned llms and diffusion models." ICLR 2024.

---

> > ### Comment · Reviewer_df7d · 2025-08-06
> >
> > The reviewer thanks the authors for their detailed response and has carefully reviewed the provided replies. While the new computational feasibility results are appreciated, several aspects would benefit from further clarification.
> >
> > In the 'Scalability to Different Dataset Sizes' table, results are reported for three dataset sizes. For the two larger sizes, both DataInf and TracIn runtimes are included. However, for the smallest size (30×5), only the DataInf runtime is reported, and the corresponding TracIn result appears to be missing. As a result, the table contains five rows of data rather than the expected six, assuming complete coverage across methods and dataset sizes.
> >
> > Similarly, in the 'Scalability to Different Model Sizes' table, a consistent presentation would include four rows—two for each model, covering both DataInf and TracIn. However, the table contains an additional fifth row that reports a TracIn runtime without any associated model information or a corresponding DataInf result.
> >
> > The reviewer would appreciate clarification on these discrepancies.

---

> > > ### Author Response · Authors · 2025-08-06
> > > **Reply to Response**
> > >
> > > We apologize for the discrepancy, there seems to have been an error while copy-pasting the tables, here are the actual tables:
> > >
> > >
> > > * **Different Dataset Sizes**:
> > >
> > > | Scalability to different Dataset Sizes | Method  | Runtime (seconds) |
> > > | -------------------- | ------- | --------------------------- |
> > > | Mistral-7B-v0.1, 3000×500         | DataInf | 163,721                     |
> > > |                      | TracIn  |  0.0135    |
> > > | Mistral-7B-v0.1, 300×50           | DataInf | 402.77                      |
> > > |                      | TracIn  | 3.314×10^-5                |
> > > | Mistral-7B-v0.1, 30×5             | DataInf | 15.3                        |
> > > |                      | TracIn  | 1.192×10^-6                |
> > >
> > >
> > > * **Different model sizes:**
> > >
> > > | Scalability to different Model Sizes | Method  | Runtime (seconds) |
> > > | -------------------- | ------- | --------------------------- |
> > > | Qwen2.5‑3B, 300×50   | DataInf | 58.063                      |
> > > |                      | TracIn  | 2.789×10^-5                 |
> > > | Qwen2.5‑32B, 300×50  | DataInf | 2,533.46                    |
> > > |                      | TracIn  | 5.245×10^-6                 |
> > >
> > >
> > >
> > > We hope this provides greater clarity.

---

> > > > ### Comment · Reviewer_df7d · 2025-08-07
> > > >
> > > > The reviewer thanks the authors for the prompt reply. Further clarification on the following point would be appreciated.
> > > >
> > > > Specifically, the reported runtimes for the TracIn method appear counter-intuitive with respect to model scaling. The tables indicate that for a dataset of size 300×50, the runtime for the 7B parameter Mistral model is $3.314\times10^{-5}$ seconds. However, for the significantly larger 32B parameter Qwen2.5 model, the reported runtime paradoxically decreases to $2.789\times10^{-5}$ seconds.
> > > >
> > > > This suggests that whilst the model size increased by over fourfold, the computation time unexpectedly fell. An explanation for this result would strengthen the reviewer's trust in the results.

---

> > > > > ### Author Response · Authors · 2025-08-07
> > > > > **Reply to Comment**
> > > > >
> > > > > **Further Clarification on TracIn runtime:**
> > > > >
> > > > > We thank the reviewer for this excellent observation. We were investigating this counter-intuitive result when our university compute cluster experienced an outage today. However, we've identified the source of the discrepancy.
> > > > >
> > > > > We believe the paradoxical TracIn runtime is likely due to different numerical precisions used specifically for Qwen2.5-32B:
> > > > >
> > > > > - Qwen2.5-3B: Computed in FP32
> > > > > - Qwen2.5-32B: Computed in FP16 to fit on 4 Ada 6000 GPUs
> > > > >
> > > > > The FP16 operations on the 32B model are 2-4× faster on modern GPUs [1], which explains why despite having much more parameters, the TracIn computation appears faster (note that it only relies on the inner product of gradients so it is a fast matrix multiplication). In contrast, DataInf scales as expected (58s → 2,533s) because it relies on the complex inverse Hessian-vector product which dominates the runtime, and the discrepancy in precision did not play a major role.
> > > > >
> > > > > We apologize for this inconsideration on our part. Owing to the reviewer's excellent observation we spent time trying to figure this out. If desired, we can try to re-run Qwen2.5-32B TracIn with FP32 once we are told that the servers are back up, on a larger 8 GPU machine.
> > > > >
> > > > > Furthermore, we would like to emphasize that this discrepancy in precision does not impact our findings:
> > > > > - All of our results in the paper are using the DataInf computation with full precision used everywhere.
> > > > > - TracIn runtimes were included only for completeness.
> > > > > - The precision difference applies only to the Qwen2.5-32B runtimes.
> > > > >
> > > > > We appreciate the reviewer's careful attention to these details and hope this clarification helps.
> > > > >
> > > > > ---
> > > > >
> > > > > References:
> > > > >
> > > > > [1]. Micikevicius, Paulius, et al. "Mixed precision training." arXiv preprint arXiv:1710.03740 (2017).

---

> > > > > > ### Comment · Reviewer_df7d · 2025-08-08
> > > > > >
> > > > > > The reviewer thanks the authors for their response. If possible, it would be great to re-run the Qwen2.5-32B TracIn with FP32 and share the updated results with the reviewer.

---

> > > > > > > ### Author Response · Authors · 2025-08-09
> > > > > > > **Updated results**
> > > > > > >
> > > > > > > Here are the new results. We appreciate engaging with the reviewer and hope that we clarified the concerns to warrant an increase in score.
> > > > > > >
> > > > > > > | Scalability to different Model Sizes | Method  | Runtime (seconds) |
> > > > > > > | -------------------- | ------- | --------------------------- |
> > > > > > > | Qwen2.5‑32B, 300×50  | DataInf | 2,824.16                    |
> > > > > > > |                      | TracIn  | 6.823605×10^−5                 |

---

> > > > > > > > ### Comment · Reviewer_df7d · 2025-08-09
> > > > > > > >
> > > > > > > > The reviewer’s concerns have been addressed. The reviewer appreciates the authors’ efforts to improve the work and will raise the score to 4.

---

> > > > > > > > > ### Author Response · Authors · 2025-08-09
> > > > > > > > >
> > > > > > > > > Thank you for helping strengthen our work; we appreciate it. We also thank the reviewer for increasing their score to 4.

---

### Official Review · Reviewer_8TjZ · 2025-07-03

**Clarity:** 3
**Significance:** 3
**Originality:** 3
**Rating:** 5
**Confidence:** 4

**Summary:**

This paper introduces LayerIF, a novel framework for evaluating the quality of individual layers within Large Language Models (LLMs). The main idea is to adapt Influence Functions (IFs), a technique used to measure a single training data point's effect on model predictions, to determine the importance of each model layer.

Unlike other methods which use model-centric metrics (like weight magnitudes), LayerIF gives a data-driven measure of layer quality.  The main computation involves determining the sensitivity to a given layers parameters to the training data, which involves calculating the gradient and Hessian per layer (or transformer block).

The authors test their method in two applications: (1) expert allocation in MoE models, where the method allocates experts based on the influence/importance score instead of standard uniform allocation and (2) layer-wise pruning, which prunes layers with higher influence scores.

**Questions:**

I alluded to my questions in the strengths/weaknesses but will state them explicitly here:
1. Could you provide a more quantitative analysis of why Hessian-free methods like TracIn fail to produce useful layer-quality estimates? For example, in the Hessians the authors compute, how badly does the Gauss-Newton approximation fail?
2.  Can they add in some discussion about the computational overhead?  For example, for a 7B parameter model on a typical GLUE task, what is the approximate wall-clock time and memory required to compute the scores?
3.  There is an interesting direction for multimodal applications.  Currently, in many vision MoEs, there is really just an explicit routing to vision experts.  It would be very interesting if this method could be applied to get VLAs to perform better or if in those cases, it is always the case to explicitly route to vision experts.  Have the authors thought about how this might be used to set up training in those sorts of applications?

**Ethical Concerns:**

["NO or VERY MINOR ethics concerns only"]

**Final Justification:**

framework for evaluating the quality of individual layers within LLMs. The main idea is to adapt Influence Functions (IFs), a technique used to measure a single training data point's effect on model predictions, to determine the importance of each model layer.

While the applications are somewhat limited due to the high computational complexity of evaluating the influence functions, which in my opinion remains the primary weakness of this work, the experiments are thorough and one could envision some downstream applications to multimodal MoEs.  The authors also addressed my questions about if a Gauss-Newton approximation could perform as well (for lower computational cost) through additional experiments.

Therefore I am going with a decision of 'accept.'

**Limitations:**

yes

**Quality:**

3

**Strengths And Weaknesses:**

Strengths:

* Applying influence functions to determine the importance of architectural components / layers instead of data points is a clever extension of an existing method.  The idea of generating different layer-importance rankings depending on downstream tasks makes intuitive sense over single allocation strategies like LoRA variants.
* The authors provide a thorough empirical evaluation of their method. The consistent, though modest, performance gains across two different applications (MoE and pruning), multiple LLMs (Mistral-7B, Gemma-7B), and various experimental settings demonstrate the potential of their method.

Weaknesses:

* There are no major weaknesses.  The primary areas of improvement relate to the clarity of certain methodological choices and the practical implications of the method's computational cost.
  * for example, the explanation of how raw LayerIF scores are mapped to expert scores could benefit from a bit more explanation and is quite dense.  It is not immediately clear how the total target sum $T$ is determined and why the specific power transformation and normalization steps were selected.  Seeing an explicit example here would be helpful.  Also, why are the raw scores noisy between adjacent layers?
* The use of a Hessian-based method (DataInf), while efficient for LoRA, is a computational hurdle for the broader applicability of this method to large models perhaps in MoE routing during pretraining.  It is interesting that approximations like TracIn perform poorly for the task. Is the Gauss-Newton form a bad approximation of the true Hessian in this case?
   * Including some numbers here would be useful, both for wall-clock time to compute the influence function for different model sizes, as well as how it scales as a function of the data points involved.
   * It would be interesting to incorporate this method during pretraining, perhaps periodically updating the probabilities with which experts are routed, but there may be significant computational bottlenecks in that setting.  Otherwise, the broader applications of the method seem more limited to smaller-scale, finetuning setups.

---

> ### Author Rebuttal · Authors · 2025-07-31
>
> We are grateful to the reviewer for their detailed feedback of our work, and are appreciative of their time, efforts, and consideration. Below we provide more discussion on some of the points raised.
>
> * **Explicit Walk-through of the algorithm**:
>
>     We thank the reviewer for their suggestion and will improve the clarity of the methods (as well as design choices) as requested. We will aim to provide a concrete step-by-step walk-through of our method, in the revised version of the paper, as described below:
>
>     **Concrete walk‑through for expert allocation (CommonQ dataset, 32‑layer LLM, T=160 experts, beta=3)**
>
>     Below we provide the exact numbers produced by our implementation.
>     For brevity only the first 5 layers are shown; the remaining layers follow identically.
>
>     | Layer $i$ | Raw IF score $v_i$ | Inverted $ṽ_i$ | Power‑scaled $v̂_i$ | Fractional $f_i$ | Final experts $s_i$ |
>     | --------- | --------------------------------------- | ------------------------------------------------------------- | -------------------------------------------------------------- | ------------------------ | --------------------------- |
>     | 0         | -5.01×10¹¹                            | 5.01×10¹¹                                                        | 1.26×10³⁵                                                              | 	1.12                    | **2**                       |
>     | 1         | -8.36×10¹¹                            | 8.36×10¹¹                                                    | 5.84×10³⁵                                                    | 5.21                    | **6**                       |
>     | 2         | -7.99×10¹¹                            | 7.99×10¹¹                                                   | 5.10×10³⁵                                                    | 4.55                     | **6**                       |
>     | 3         | -8.46×10¹¹                            | 8.46×10¹¹                                                    | 6.05×10³⁵                                                    | 5.40                     | **6**                       |
>     | 4         | -8.16×10¹¹                            | 8.16×10¹¹                                                   | 5.43×10³⁵                                                    | 4.84                     | **6**                       |
>     | …         | …                                       | …                                                             | …                                                              | …                        | …                           |
>
>
>
>   (1) **Step1 (sign inversion).** Invert the negative values to positive.
>
>   (2) **Step2 (power transform).** Amplify disparities by raising to the power beta to get v̂.
>
>   (3) **Step3 (normalise to budget).** Scale v̂ so that the fractional allocations ∑fi equal **T−m=128** extra experts (m=32 layers).
>
>   (4) **Step4 (floor+redistribute).** Set si=⌊fi⌋+1, guaranteeing ≥1 expert/layer, then distribute the remaining 24 units to the layers with the largest fractional parts.
>
>   Finally, the complete allocation for **CommonQ** becomes:
>
>     ```[2, 6, 6, 6, 6, 5, 7, 6, 8, 7, 6, 7, 6, 6, 6, 5, 6, 6, 5, 4, 5, 5, 4, 2, 4, 5, 3, 4, 2, 3, 4, 3]```
>
>     **Interpretation.** The middle layers receive the most experts, indicating they are comparatively under‑trained for CommonQ; the first and very deep layers receive lower number of experts, reflecting higher estimated quality.
>
>     Similarly we will also provide an example for the layer-wise sparsity allocation setting. If the reviewer would like us to take any other steps to improve clarity, we are more than happy to do so.
>
> ___
> ___
>
> * **Why are the raw scores noisy between adjacent layers?**:
>
>     We thank the Reviewer for this important question. Influence‑function estimates at adjacent layers are inevitably noisy: they are computed from stochastic Hessian approximations and mini‑batch gradients, and thus contain high‑frequency fluctuations  that can misrepresent the true differences in layer quality. Grosse et al. [1] visualize this variability directly: the layer × token heatmap in Figure 4 (p.20) reveals sharp red/teal oscillations across adjacent layers, reflecting high-frequency fluctuations in estimated influence. The authors also note in Section 2.1 that influence values are *extremely heavy-tailed*, which complicates traditional statistical analysis and motivates the need for robust aggregation strategies.
>
>     Hence, we apply a Savitzky–Golay smoother (window = 5, poly‑order = 2) to: a) Suppress spurious high‑frequency noise, while b) Preserving low‑frequency structure, the monotonic rise–fall pattern that truly indicates which layers of the LLM are under or over‑trained.
>
>     We will add a sentence clarifying this rationale in Section 3.4 accordingly.
>
> ___
> ___
>
> * **Wall Clock times and Memory Consumption for DataInf and TracIn under varying datasets and model sizes:**
>
>     We thank the reviewer for raising the important point regarding the computational times of Influence Function methods. Below, we provide wall-clock timings for computing the influence function for a single layer across varying training and validation set sizes. We used 4× NVIDIA RTX 6000 Ada Generation GPUs to perform these experiments.
>
>
>     | Scalability to different Dataset Sizes | Method  | Runtime (seconds) |
>     | -------------------- | ------- | --------------------------- |
>     | Mistral-7B-v0.1, 3000×500         | DataInf | 163,721                     |
>     |                      | TracIn  |  0.0135    |
>     | Mistral-7B-v0.1, 300×50           | DataInf | 402.77                      |
>     |                      | TracIn  | 3.314×10^-5                |
>     | Mistral-7B-v0.1, 30×5             | DataInf | 15.3                        |
>
>
>     As we can see the time increases non-linearly with larger datasets, but we intend to clarify that in our experiments, having approximately 300-500 training samples and 50-100 validation samples is typically sufficient to gauge how well an LLM layer generalizes to a task. Moreover, the current implementation of DataInf runs primarily on CPU; preliminary tests show that offloading key computations to the GPU yields up to a 2× speedup, indicating substantial room for further efficiency improvements.
>
>     Similarly, we report the runtime for computing the influence function for 1 layer for different model sizes.
>
>
>     | Scalability to different Model Sizes | Method  | Runtime (seconds) |
>     | -------------------- | ------- | --------------------------- |
>     |                      | TracIn  | 1.192×10^-6                |
>     | Qwen2.5‑3B, 300×50   | DataInf | 58.063                      |
>     |                      | TracIn  | 2.789×10^-5                 |
>     | Qwen2.5‑32B, 300×50  | DataInf | 2,533.46                    |
>     |                      | TracIn  | 5.245×10^-6                 |
>
>
>     As expected, larger models incur higher influence computation times; however, these costs are     negligible relative to the overall expense of pretraining large language models. For context, training     LLaMA2‑70B required approximately 1.72 million GPU hours. By comparison, our method     represents a minor overhead and can be integrated into the broader pretraining pipeline to diagnose     underperforming layers and nudge the model toward more uniform training dynamics. This can     support developers in identifying and addressing layer‑specific inefficiencies during model     development.
>
>
>     Finally, the peak GPU memory usage during influence computation for a dataset with 300 training and 50 validation samples was approximately 35,000 MiB for a 7B parameter model. This footprint is relatively modest given the model size and suggests that influence-based diagnostics can be feasibly incorporated into GPU-based training pipelines without exceeding typical memory constraints.
>
> ___
> ___
>
> * **Gauss-Newton Approximation's applicability:**
>
>     The main bottleneck with the Gauss-Newton Hessian is the time required for IF computation. While doing initial experiments, we considered utilizing the Guass-Newton Hessian instead of the DataInf approximation. However, even the time taken to compute the **Gauss-Newton Hessian IF** for 30 Train and 5 Validation samples was **2494.72 seconds** for a 7B model whereas the equivalent experiment with the **DataInf IF** approximation took only **15.3 seconds**. Hence, it made it unfeasable for our experiments, which required 300 Train and 50 Validation samples.
>
> ___
> ___
>
> * **Applicability VLA settings:**
>
>     Thank you. We appreciate the reviewer’s suggestion and agree that multimodal mixtures‑of‑experts (MoEs) and particularly VLAs represent an exciting next step. There is a conceptual fit as our influence‑based layer‑quality metric is modality‑agnostic: it only requires gradients and Hessian–vector products for the parameters under consideration. In principle, we can therefore:
>
>     1. **Compute per‑expert influence** for vision, language, and cross‑modal experts separately;
>     2. **Compare their sensitivity profiles** on joint vision–language objectives (e.g., image‑text contrastive loss, VQA loss);
>     3. **Adapt the routing policy** according to these influence scores.
>
>     We have actually begun conducting some preliminary research in this very exciting area as part of follow-up work, and will be releasing it in the near future for the community.
>
>
> ---
> ___
>
> Thank you once again for all your efforts in helping to strengthen our work. We hope that our comments helped address your concerns and if so, would be extremely obliged if you could consider increasing your score.
>
> ___
> ---
>
> References:
>
> [1] Grosse, Roger, et al. "Studying large language model generalization with influence functions." arXiv preprint arXiv:2308.03296 (2023).

---

> ### Author Response · Authors · 2025-08-03
> **Additional Results on Gauss-Newton Hessian.**
>
> For the sake of completeness, we conducted an additional experiment to assess the effectiveness of the Gauss-Newton Hessian on the *Text_Science_Q* dataset using the Mistral-7B-v0.1 model. Due to  time constraints, layer-wise influence scores were computed on a subset of 30 training and 5 validation samples. The resulting model achieved an accuracy of **72.57%**, which is substantially lower than that of the DataInf-based methods.

---

> > ### Comment · Reviewer_8TjZ · 2025-08-04
> > **Acknowledgment of author response**
> >
> > I thank the authors for their responses to my questions and additional clarifications.  I look forward to the updated draft.
> >
> > I have no further questions, and will update my score.

---

> > > ### Author Response · Authors · 2025-08-09
> > > **Reply to response**
> > >
> > > We wholeheartedly thank the reviewer for engaging with us and helping us in strengthening our work.

---

### Official Review · Reviewer_w8JJ · 2025-07-14

**Clarity:** 2
**Significance:** 3
**Originality:** 3
**Rating:** 4
**Confidence:** 3

**Summary:**

This paper studies how to adopt the classical influence functions (IF) to estimate the importance of each layer in LLM. The paper proposes LayerIF and shows how LayerIF can guide:
- expert-count allocation in mixture-of-experts (MoE) training
- layer-wise sparsity pruning.

Experiments on a suite of GLUE and commonsense benchmarks demonstrate modest but consistent gains over existing heuristics.  Because LayerIF scores are computed offline, the method does not add inference-time latency.

**Questions:**

See Weakness and Questions.

**Ethical Concerns:**

["NO or VERY MINOR ethics concerns only"]

**Final Justification:**

My primary concern is about the principles behind the three variants: TOP‑25%, +VE, and ALL. The authors's rebuttal has addressed my concern.

**Limitations:**

Yes.

**Quality:**

2

**Strengths And Weaknesses:**

## Strength
1. The work is the first to leverage IF at layer granularity in LLMs, providing new insight into module-level contributions and may potentially inspire more principled schemes for MoE routing and structured pruning.

2. IF computation is traditionally expensive. However, in LayerIF, computation of IF only occurs once during analysis or training.  Both the proposed MoE allocation and pruning methods rely on pre-computed IF scores, and the computation in the inference stage remains unchanged.

## Weakness and Questions

1. The technical presentation in Sections 3.3 and 3.4 is very difficult to follow. For example, Lines 204–232 interleave motivations, notations, math expressions, and constraints in long paragraphs, without the aid of any diagram or algorithm block.

2. The method description is inconsistent. Section 3.3 states that negative-influence samples are excluded when aggregating IF values. However, Section 4.1 evaluates three variants, ALL, +VE, and TOP-25 %.

3. Table 1 shows that the best-performing variant differs across tasks (e.g., +VE for OpenbookQA, ALL for CoLA, TOP-25 % for MRPC).  A principled guideline for choosing the variants or an automatic tuning strategy is missing.

4. The paper claims that “IFs measure the sensitivity of model loss to individual training points and the loss of well-trained layers should show smaller changes”. This is a bit counterintuitive, removing an important layer should increase loss significantly.

---

> ### Author Rebuttal · Authors · 2025-07-31
>
> We are grateful to the reviewer for their feedback of our work, and are appreciative of their time, effort, and consideration. Below we provide more discussion on the points raised.
>
> * **Technical Presentation/Clarity**:
>
>     We thank the reviewer for their suggestion and will improve the clarity of Section 3.3 and Section 3.4 as requested. We will aim to provide a concrete step-by-step walk-through of our method, in the revised version of the paper, as described below:
>
>
>     **Concrete walk‑through for expert allocation (CommonQ dataset, 32‑layer LLM, T=160 experts, beta=3)**
>
>     Below we provide the exact numbers produced by our implementation.
>     For brevity only the first 5 layers are shown; the remaining layers follow identically.
>
>     | Layer $i$ | Raw IF score $v_i$ | Inverted $ṽ_i$ | Power‑scaled $v̂_i$ | Fractional $f_i$ | Final experts $s_i$ |
>     | --------- | --------------------------------------- | ------------------------------------------------------------- | -------------------------------------------------------------- | ------------------------ | --------------------------- |
>     | 0         | -5.01×10¹¹                            | 5.01×10¹¹                                                        | 1.26×10³⁵                                                              | 	1.12                    | **2**                       |
>     | 1         | -8.36×10¹¹                            | 8.36×10¹¹                                                    | 5.84×10³⁵                                                    | 5.21                    | **6**                       |
>     | 2         | -7.99×10¹¹                            | 7.99×10¹¹                                                   | 5.10×10³⁵                                                    | 4.55                     | **6**                       |
>     | 3         | -8.46×10¹¹                            | 8.46×10¹¹                                                    | 6.05×10³⁵                                                    | 5.40                     | **6**                       |
>     | 4         | -8.16×10¹¹                            | 8.16×10¹¹                                                   | 5.43×10³⁵                                                    | 4.84                     | **6**                       |
>     | …         | …                                       | …                                                             | …                                                              | …                        | …                           |
>
>
>
>   (1) **Step1 (sign inversion).** Invert the negative values to positive.
>
>   (2) **Step2 (power transform).** Amplify disparities by raising to the power beta to get v̂.
>
>   (3) **Step3 (normalise to budget).** Scale v̂ so that the fractional allocations ∑fi equal **T−m=128** extra experts (m=32 layers).
>
>   (4) **Step4 (floor+redistribute).** Set si=⌊fi⌋+1, guaranteeing ≥1 expert/layer, then distribute the remaining 24 units to the layers with the largest fractional parts.
>
>   Finally, the complete allocation for **CommonQ** becomes:
>
>     ```[2, 6, 6, 6, 6, 5, 7, 6, 8, 7, 6, 7, 6, 6, 6, 5, 6, 6, 5, 4, 5, 5, 4, 2, 4, 5, 3, 4, 2, 3, 4, 3]```
>
>     **Interpretation.** The middle layers receive the most experts, indicating they are comparatively under‑trained for CommonQ; the first and very deep layers receive lower number of experts, reflecting higher estimated quality.
>
>     Similarly we will also provide an example for the layer-wise sparsity allocation setting. If the reviewer would like us to take any other steps to improve clarity, we are more than happy to do so.
>
> ___
> ___
>
> * **The method description is inconsistent in Section 3.3**:
>
>
>     We appreciate the reviewer’s attention to this apparent inconsistency and apologize for the confusion. We intend to clarify that Section 3.3 describes the default pipeline used throughout the paper. Specifically, we aggregate only positive‑influence samples (`+VE`) and discard negative ones.
>
>     In Section 4.1, we include two additional variants, `ALL`, which retains every sample, and `TOP 25%`, which selects the top quartile of positive IF scores by magnitude, as part of a controlled ablation study. These ablations were included at this stage because including them earlier would have introduced unnecessary complexity and we wanted to empirically test our design choices.
>
>     These variants were included solely to assess robustness and justify our aggregation choice; they were not used in the main experimental pipeline, as reflected in Figures 2 and 3 and Table 2.
>
>     Additionally, we perform another ablation in the rebuttal period where we compare keeping all of the samples against keeping only the positive samples on the Gemma-7b model.
>
>     | Dataset            | IF (+ve)  | IF (ALL) |
>     |--------------------|------------------|-----|
>     | MRPC               | 84.75            | 82.49 |
>     | CoLA               | 86.96            | 85.14 |
>     | Common_Q           | 83.46            | 82.06 |
>     | Openbook_Q         | 88.20            | 86.20 |
>     | Text_Science_Q     | 93.66            | 94.56 |
>     | **Average**        | **87.81**        | **86.09** |
>
> &nbsp;&nbsp;&nbsp;&nbsp;&nbsp;&nbsp;&nbsp;&nbsp;Here, we see that IF (+ve) was significantly better performing on average and individually more performant on 4 out of the 5 datasets.
>
> ___
> ___
>
> * **A principled guideline for choosing the variants or an automatic tuning strategy is missing:**
>
>
>     We thank the reviewer for highlighting this point. We would like to point that while Table 1 shows some variation across tasks as the reviewer mentioned e.g., `+VE` performs best on OpenbookQA, and `TOP‑25%` on MRPC and Text Science Q, the overall trend strongly favors the IF based methods, particularly `TOP‑25%`. Specifically, `TOP‑25%` achieves:
>
>     - the highest **macro-average** performance (82.27%) across all tasks,
>     - has the lowest standard deviation 1.84.
>
>     This robustness makes `TOP‑25%` a strong default in practice. The use of a fixed, global aggregation threshold is also consistent with precedent in prior influence function research. For instance, in [1], as evident in Table 5 and Table 6, aggregate performance trends across tasks are generally more informative than per-dataset maxima. This is also evident in our results.
>
>     However, we agree with the reviewer that dataset-specific optimal thresholding might yield further gains, and will definitely aim to study that exploration more in follow up work.
>
> ___
> ___
>
> * **Removing an important layer should increase loss significantly:**
>
>     We believe there may be a misunderstanding and would like to clarify our intended meaning. When we state in the Introduction that "IFs measure the sensitivity of model loss to individual training points and the loss of well-trained layers should show smaller changes," we are emphasizing that well-trained layers tend to be more stable i.e., they exhibit lower sensitivity to perturbations in the training data. This interpretation is further elaborated in the “Justification” paragraph of Section 3.2.
>
>     We fully agree with the reviewer that removing important layers would significantly degrade model performance and increase loss. Our empirical results support this: the layers identified by our method as less sensitive (and thus likely better trained) receive fewer LoRA experts and are pruned more conservatively, leading to stronger downstream performance.
>
>     To further emphasize this point, we conduct an ablation where we reversed our methodology by pruning the less sensitive layers more at a 70% sparsity budget on Mistral-7b-v0.1. We present the results below.
>
>
>
>     |             Method  | Magnitude | Wanda  | SparseGPT | Average  |
>     |---------------------|-----------|--------|-----------|----------|
>     | Our Method          | 33.439    | 32.497 | 41.143    | 35.693   |
>     | Reversed Allocation | 33.101    | 32.015 | 38.450    | 34.522   |
>
> &nbsp;&nbsp;&nbsp;&nbsp;&nbsp;&nbsp;&nbsp;&nbsp;We will add this ablation to the revised version of our paper.
>
> ---
> ___
>
> Thank you again for taking the time to review our paper. We hope that our responses alleviated the concerns raised, and would be happy to engage further during the discussion phase in case there are any others remaining.
>
> ---
> ___
>
> References:
>
> [1] Chhabra, Anshuman, et al. "Outlier Gradient Analysis: Efficiently Identifying Detrimental Training Samples for Deep Learning Models." ICML 2025.

---

> ### Comment · Reviewer_w8JJ · 2025-08-05
>
> Thank you for your response. I will consider increasing my score. (It seems that I can't temporarily modify the score at this stage; Once I modify, it would be my final score.)
>
> Though the authors provide empirical evidence that TOP‑25% is strongest, why selecting the top quartile of positive IF scores is better than +VE and ALL still needs more justification. In principle, why does selecting partial positive IF help? And how can we determine an appropriate ratio?

---

> > ### Author Response · Authors · 2025-08-05
> > **Response to comment**
> >
> > We sincerely thank the reviewer for their constructive engagement and for considering increasing their score. We appreciate the opportunity to provide deeper justification for our aggregation strategy.
> >
> > ## Principled Justification for TOP-25\% Selection
> >
> > The reviewer raises an excellent question about *why* selecting the top quartile of positive IF scores outperforms both +VE (all positive) and ALL variants. We believe there are several principled reasons:
> >
> > ### 1. **Signal-to-Noise Ratio Optimization**
> > While positive/negative IF scores indicate training samples that increase/reduce validation loss, not all scores are equally informative. The TOP-25\% selection acts as a **noise filter** that captures the most salient training-validation relationships:
> >
> > - **High-magnitude scores** represent training samples with strong, consistent effects on validation performance
> > - **Low-magnitude scores** often reflect spurious correlations or marginal contributions that may introduce noise
> > - By focusing on the top quartile, we are likely improving estimation accuracy of each layer's quality by isolating training patterns it has most successfully learned to leverage
> >
> > ### 2. **Theoretical Connection to Robust Statistics**
> > Our TOP-25\% approach aligns with principles from robust statistics, where trimmed means and percentile-based estimators are known to be more stable than full-sample statistics [1,2]. In the context of influence functions:
> >
> > - The full +VE aggregation can be dominated by many small, potentially noisy positive values
> > - The TOP-25\% acts as a **trimmed estimator** that is more robust to distributional assumptions
> > - This is particularly important given that IF approximations in deep networks can be noisy, especially for small influence values [3]
> >
> >
> > ### 3. **Heavy-Tailed Distributions**
> > In our analysis, we observed that positive IF scores follow a heavy-tailed distribution across multiple datasets and models, this is also observed in other IF literature [4]. This concentration of influence in the upper tail suggests that most of the meaningful signal resides in the top-scoring samples, making TOP-25\% a natural cutoff point. The heavy-tailed nature means that a small fraction of training samples account for a disproportionately large portion of the total positive influence, while the majority contribute only marginally.
> >
> > ## Determining the Appropriate Ratio
> >
> > The reviewer correctly asks how to determine the appropriate selection ratio. While we empirically found 25\% to be effective, we propose several strategies for principled selection:
> >
> > ### **Elbow Method Analysis**
> > We can identify the natural cutoff by examining the cumulative influence curve:
> > - Sort positive IF values in descending order
> > - Plot cumulative influence vs. percentage of samples
> > - The "elbow" point where marginal contribution drops significantly provides a data-driven threshold.
> >
> > Additionally, cross-validation on a development set can help determine task-specific optimal thresholds, and future work could explore information-theoretic criteria for automatic threshold selection.
> >
> > ## Additional Experimental Evidence
> >
> > Beyond the results provided in our initial response, we conducted additional ablations during the rebuttal period on Gemma-7b, which we provide below:
> >
> > | Dataset            | IF (+ve)  | IF (ALL) |
> > |--------------------|-----------|----------|
> > | MRPC               | 84.75     | 82.49    |
> > | CoLA               | 86.96     | 85.14    |
> > | Common_Q           | 83.46     | 82.06    |
> > | Openbook_Q         | 88.20     | 86.20    |
> > | Text_Science_Q     | 93.66     | 94.56    |
> > | **Average**        | **87.81** | **86.09**|
> >
> > Here, we see that IF (+ve) was significantly better performing on average and individually more performant on 4 out of 5 datasets. Which shows the benefit of aggregating the positive samples.
> >
> >
> > We believe these analyses significantly strengthen our understanding of why positive and partial positive IF selection is beneficial and will incorporate these insights into our revised manuscript.
> >
> > ---
> >
> > Thank you again for your thoughtful engagement with our work. We hope this expanded justification addresses your concerns about the principled basis for our aggregation strategy. We would be happy to discuss any additional aspects of our methodology.
> >
> >
> > ---
> >
> > References:
> >
> > [1] Huber, P. J. (1981). Robust statistics. John Wiley & SonsRobust statistics. John Wiley & SonsRobust statistics. John Wiley & Sons.
> >
> > [2] Wilcox, R. R. (2011). Introduction to robust estimation and hypothesis testing. Academic press.
> >
> > [3] Basu et al. (2021). Influence Functions in Deep Learning Are Fragile. ICLR 2021.
> >
> > [4] Grosse, Roger, et al. "Studying large language model generalization with influence functions." arXiv preprint arXiv:2308.03296 (2023).

---

> > > ### Author Response · Authors · 2025-08-09
> > > **Reply to comment**
> > >
> > > The authors would like to thank the reviewer for engaging with us and helping us in strengthening our work. We appreciate that they are planning to increase their score and would like to clarify any concerns that remain.

---

### Decision · Program_Chairs · 2025-09-17

**Decision:**

Accept (poster)

**Comment:**

The paper presents a novel approach of using influence functions to estimate the importance of different layers in a large language model (LLM). Although the idea of influence functions is not new, its application to LLMs as done in this paper is novel. Some applications of the proposed LayerIF approach, including expert allocation and pruning, have been discussed and experimented in this work too, which illustrates the usefulness of LayerIF in practice. I believe that LayerIF provides a new perspective for a more systematic understanding of LLMs, which is a very important problem.

The reviewers had some comments especially about the clarity of some parts in the paper and experimental results. After discussions, all the reviewers recommended accepting this paper.